# Top-down feedback controls spatial summation and response amplitude in primate visual cortex

Lauri Nurminen[1], Sam Merlin[1,2], Maryam Bijanzadeh[1,3], Frederick Federer[1] & Alessandra Angelucci[1]

Sensory information travels along feedforward connections through a hierarchy of cortical areas, which, in turn, send feedback connections to lower-order areas. Feedback has been implicated in attention, expectation, and sensory context, but the mechanisms underlying these diverse feedback functions are unknown. Using specific optogenetic inactivation of feedback connections from the secondary visual area (V2), we show how feedback affects neural responses in the primate primary visual cortex (V1). Reducing feedback activity increases V1 cells' receptive field (RF) size, decreases their responses to stimuli confined to the RF, and increases their responses to stimuli extending into the proximal surround, therefore reducing surround suppression. Moreover, stronger reduction of V2 feedback activity leads to progressive increase in RF size and decrease in response amplitude, an effect predicted by a recurrent network model. Our results indicate that feedback modulates RF size, surround suppression and response amplitude, similar to the modulatory effects of visual spatial attention.

[1] Department of Ophthalmology and Visual Science, Moran Eye Institute, University of Utah, 65 Mario Capecchi Drive, Salt Lake City, UT 84132, USA. [2] Present address: Medical Science, School of Science and Health, Western Sydney University, Campbelltown, NSW 2560, Australia. [3] Present address: Department of Neurological Surgery, UCSF, San Francisco, CA 94143, USA. These authors contributed equally: Lauri Nurminen, Sam Merlin. Correspondence and requests for materials should be addressed to A.A. (email: alessandra.angelucci@hsc.utah.edu)

In addition to well-studied bottom-up feedforward inputs, the visual cortex receives a much denser network of feedback inputs from higher-order cortical areas[1] whose role remains hypothetical. Feedback has been implicated in several forms of top-down influences, such as attention[2,3], expectation[4] and sensory context[5,6], which affect sensory processing in diverse ways. For example, visual spatial attention, one of the most studied instances of top-down influences, has been shown to modulate neuronal response gain[2,7], surround suppression[8] and receptive field (RF) size[9]. In this study we have asked whether feedback connections can mediate such diverse effects.

To determine the cellular mechanisms underlying the influence of cortical feedback on sensory processing, we asked whether inactivating feedback from the secondary visual area (V2) alters RF size, surround suppression and response gain in the primary

visual cortex (V1). Surround suppression is the property of V1 neurons to reduce their response to stimuli inside their RF when presented with large stimuli extending into the RF surround[10–18]. This is a fundamental computation throughout the visual cortex, thought to increase the neurons' coding efficiency[19–22], to contribute to segmentation of objects boundaries[21], and to be generated by feedback connections[5,6]. However, the role of feedback in surround suppression and response gain or amplitude remains controversial. Inactivation of higher-order cortices using pharmacology, cooling or optogenetics has produced weak reduction in surround suppression in some studies[23–25], but only reduction in response amplitude in other studies[26–29]. One problem with these previous studies is that these inactivation methods suppressed activity in an entire cortical area; thus, the observed effects could have resulted from indirect pathways through the

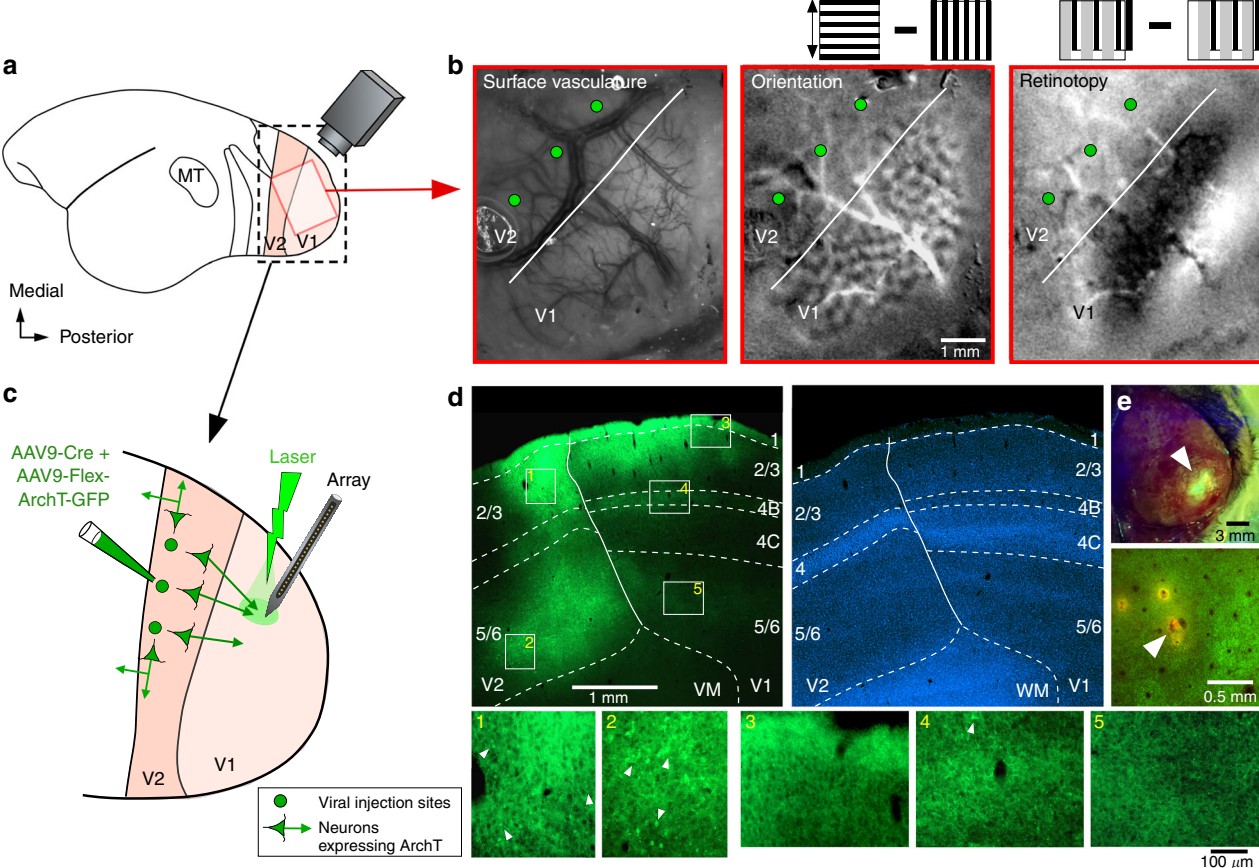

**Fig. 1** Optogenetic inactivation of V2 feedback terminals: experimental design and ArchT-GFP expression in V2 feedback terminals. **a** Schematics of the marmoset brain. Red box: approximate location of the optically imaged region in **b**. Black box: V1 and V2 region shown enlarged in **c**. **b** Optical imaging identifies V1/V2 border (white line). Left panel: cortical surface vasculature imaged under green light, used as reference to position pipettes for viral injections (green dots). Middle panel: orientation map generated by subtracting responses to 0° and 90° gratings (as shown in inset). V2 can be identified by larger orientation domains compared to V1. Right panel: retinotopic map generated by subtracting responses to 90° oriented gratings occupying complementary and adjacent strips (1° in width) of visual space (as shown in inset above; see Methods). The V1/V2 border can be identified by the presence of stripes in V1, running approximately parallel to the V1/V2 border, which are absent in V2 (as the grating parameters were optimized for V1, but not V2, cell; see Methods). **c** Schematics of the inactivation paradigm: multiple viral injections were targeted to V2, array recordings and laser photostimulation to V1. **d** ArchT-GFP expression in V1 and V2. Top left: sagittal section through V1 and V2, viewed under GFP fluorescence, showing two injection sites confined to V2, and resulting expression of ArchT-GFP in the axon terminals of V2 feedback neurons within V1 layers 1–3, 4B and 5/6 (typical feedback laminar termination pattern[32,33]). This tissue section was located near the lateralmost aspect of the hemisphere, therefore the infragranular layers are elongated due to the lateral folding-over of the cortical sheet. Solid contour: V1/V2 border. Dashed contours: laminar borders delineated on the same section counterstained with DAPI (top right). Bottom panels 1–5: higher magnification of label inside the white boxes numbered *1-5* in the top left panel. Panels 1–2 show multiple clusters of labeled somata (e.g., arrowheads) at the V2 injection sites; instead, there is only one labeled soma (arrowhead) in panel 4, and none in panels 3,5. **e** Top panel: GFP excitation (arrowhead) through the intact thinned skull, approximately two months after viral injection. Bottom panel: Tangential section through V1 showing the location of a DiI-coated electrode penetration (arrowhead) amid ArchT-GFP-expressing feedback axon terminals (green fluorescence)

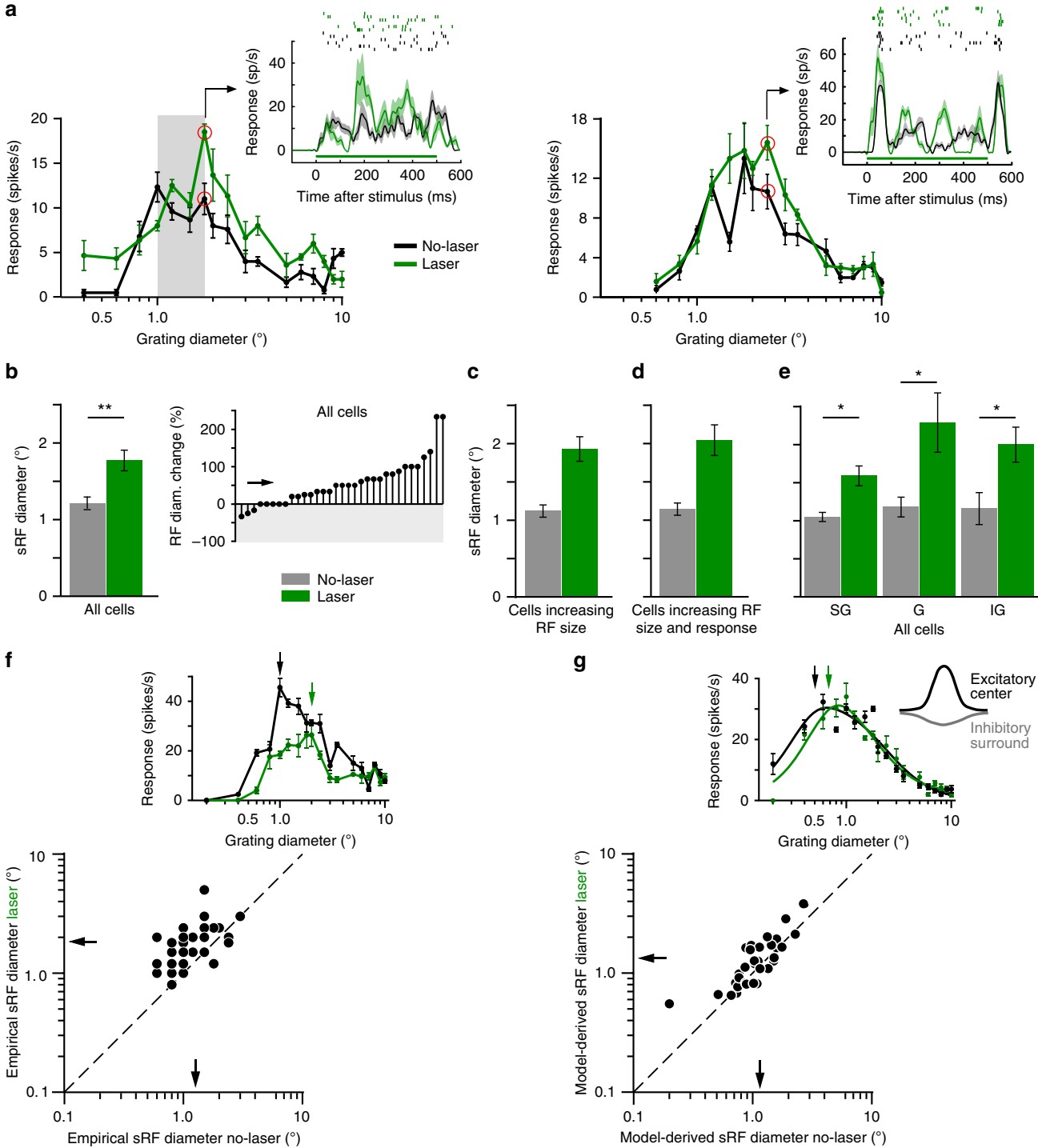

**Fig. 2** V2 feedback controls RF size. **a** Spatial summation curves for two example V1 cells recorded with (green) and without (black) laser stimulation. Gray area in left panel: proximal surround. Insets: PSTHs (Bottom; due to the smoothing filter used, response onset starts at time zero) and raster plots (Top) measured at the stimulus diameters indicated by the red circles in the respective size-tuning curves. Green horizontal line: laser-on time. Two additional example cells are shown in the insets of panels **f** and **g**. **b–e** Mean sRF size (diameter at peak response of empirically measured spatial summation curve) with and without laser stimulation for: **b** All cells (LEFT; $n = 33$); RIGHT: Cell-by-cell percent change in sRF size across the entire cell population. Downward and upward stem: decreased and increased sRF size, respectively. Arrow: mean. **c** Only cells showing increased sRF size with laser stimulation ($n = 25$; mean sRF diameter ± s.e.m. no laser vs. laser: 1.12 ± 0.08° vs. 1.93 ± 0.08°). **d** Only cells showing both increased sRF size and peak response with laser stimulation ($n = 12$; 1.14 ± 0.08° vs. 2.04 ± 0.20°). **e** Mean sRF size for all cells (as in **b**), but grouped according to layer. SG: supragranular; G: granular; IG: infra-granular. **f–g** Scatterplots of sRF diameter with and without laser stimulation for sRF diameter derived directly from the empirically measured summation curves (**f**), or from the model curves fitted to the summation data (**g**), as indicated in the insets above each scatterplot. Insets in **f** and **g** show the size tuning curve of two additional example cells. The summation data for the cell in **g** are fitted with the ROG model. Arrows in insets in **f** and **g** indicate the sRF diameters. Arrows in scatterplots: means. Dashed line in **f–g**: unity line. Here and in all remaining figures error bars are s.e.m

thalamus or other cortical areas. Moreover, these approaches did not allow fine control of inactivation levels, thus precluding potentially more physiologically relevant manipulations. To overcome the technical limitations of previous studies, we have used selective optogenetic inactivation of V2-to-V1 feedback axon terminals, rather than direct inactivation of the entire V2, while measuring spatial summation and surround suppression in V1 neurons using linear electrode arrays.

We find that V2 feedback modulates RF size, surround suppression and neuronal response amplitude in V1. As several forms of top-down influences in sensory processing have been shown to affect neuronal responses in the same way as we have shown here for feedback from V2, our study suggests that feedback connections can support a large variety of top-down effects observed in vivo.

## Results

**Specific optogenetic inactivation of feedback connections**. To express the outward proton pump Archaerhodopsin-T (ArchT)[30] in the axon terminals of V2 feedback neurons, we injected into V2 of marmoset monkeys a mixture of Cre-expressing and Cre-dependent adeno-associated virus (AAV9) carrying the genes for ArchT and green fluorescent protein (Fig. 1a, c; see Methods). This viral vector combination was used because in pilot studies we found that it produces selective anterograde infection of neurons at the injected V2 site, and virtually no retrograde infection of neurons in V1 (Fig. 1d). Intrinsic signal optical imaging was used to identify the V1/V2 border (Fig. 1a, b) and target injections to V2 (Fig. 1c, d) (see Methods). Linear array recordings were, subsequently, targeted to GFP/ArchT-expressing V1 regions (Fig. 1c, e). Trial interleaved and balanced surface laser stimulation of increasing intensity was applied to ArchT-expressing axon terminals of V2 feedback neurons at the V1 recording site (Fig. 1c; see Methods). This viral injection protocol produces ArchT-GFP expression in V2 neurons at the injected site, including neurons sending feedback projections to V1 but also other V2 neurons projecting within V2 itself or to other brain regions. However, directing the laser to V1, while shielding V2 from light, allowed us to selectively inactivate V2 feedback terminals, at least in the superficial layers of V1, leaving neurons within V2 unperturbed (Fig. 1c).

**V2 feedback affects receptive field size**. Electrophysiological recordings were performed in parafoveal V1 of anesthetized and paralyzed marmosets using 24-contact linear electrode arrays inserted orthogonal to the cortical surface, as verified by the vertical alignment of RFs and similarity of orientation preference across the array (see Methods, Supplementary Note 1 and Supplementary Fig. 1). After initial characterization of RF properties at each contact through the V1 column, we measured spatial summation curves, using drifting grating patches of increasing diameter centered on the column's aggregate RF. Typical V1 cells increase their response with stimulus diameter up to a peak (the summation receptive field, sRF, size), and are suppressed for larger stimulus sizes activating also the RF surround (Fig. 2a).

We present spatial summation measurements from 67 visually responsive and stimulus modulated, spike-sorted single units from 3 animals. Approximately 61% (41/67) of single units were significantly modulated by the laser (see Methods, for neuronal sample selection). As laser-induced heat can alter cortical spiking activity[31], we selected a safe range of laser intensities (9–43 mW/mm²), based on results from control experiments in cortex not expressing ArchT (see Supplementary Note 2 and Supplementary Figs. 2-3).

When feedback was inactivated, the majority (76%) of laser-modulated units showed a shift of the spatial summation peak towards larger stimuli, i.e., an increase in sRF size (Fig. 2); in the remainder of the cells sRF size was unchanged (15%) or decreased (9%). Moreover, in 46% of cells sRF size increase was accompanied by an increase in peak response amplitude (Fig. 2a), while in other cells peak response was decreased (e.g., Figure 2f inset) or unchanged (e.g., Figure 2g inset). This analysis was based on selecting, for each cell, the laser stimulation intensity producing the largest change in sRF size, but within the range of intensities selected on the basis of control experiments (see above and Methods) (mean irradiance across the population ± sem was $28.7 \pm 1.95$ mW/mm²). Across the entire neuronal population ($n = 33$ cells), mean sRF diameter, defined as the stimulus diameter at the peak of the empirically measured summation curve (Fig. 2f inset), was significantly smaller with intact feedback, compared to when feedback was inactivated (mean ± s.e.m: $1.27 \pm 0.10°$ vs. $1.83 \pm 0.14°$, $T$-test $p < 0.01$; Mann–Whitney U-test $p < 0.001$; see Methods), with a mean increase of $56.2 \pm 10.7\%$ ($T$-test for mean increase > 0%, $p < 0.001$; Mann–Whitney U-test, $p < 0.001$; Fig. 2b, f). Figure 2c, d illustrates the magnitude of the mean sRF size change caused by feedback inactivation, when considering only cells that showed increases in sRF size (Fig. 2c) or cells that showed increases in both sRF size and peak response magnitude (Fig. 2d).

We also examined how these changes in sRF size vary with V1 layer, as it is known that V2 feedback connections target supragranular and infragranular layers, but avoid the granular layer in V1[32,33]. We found that feedback inactivation increased mean sRF diameter in all layers (Fig. 2e) (mean ± s.e.m. no-laser vs. laser: supragranular layers $1.23 \pm 0.11°$ vs. $1.53 \pm 0.10°$; granular layer $1.31 \pm 0.17°$ vs. $2.26 \pm 0.35°$; infragranular layers $1.29 \pm 0.25°$ vs. $1.88 \pm 0.26°$; $T$-test $p < 0.05$ for all layers; Mann–Whitney U-test, $p < 0.05$ for all layers). This suggests that, at least in the granular layer, which does not receive direct feedback terminations, changes in sRF size are relayed via other layers.

Since sRF size derived from the empirically measured curves can be subject to noise, we also compared the sRF size with and without laser extracted from phenomenological model fits to the summation data, as these can provide more robust measures of sRF size. To this purpose, we fitted to the summation data two different models, namely a ratio or difference of integrals of two Gaussians (ROG or DOG model, respectively; see Methods), which have previously been shown to provide a good description of spatial summation curves in macaque V1[14,15]. In these models, a center excitatory Gaussian, corresponding to the RF center, overlaps a spatially broader inhibitory Gaussian, representing the suppressive surround (inset in Fig. 2g); the major difference between the two models is that the surround inhibits the center through division in the ROG model, but through subtraction in the DOG model (see Methods). The ROG model provided a better fit for most (79%), but not all, of the cells (see below). Therefore, we fitted both models to the spatial summation data with and without laser stimulation, and for each cell we extracted sRF size from the model that provided the best fit to that cell's data. From the fitted curve, sRF size was defined as the stimulus diameter at 95% of peak response (as in ref. [14]) (Fig. 2g inset). Importantly, we still found feedback inactivation to significantly increase sRF size when the latter was estimated from the models fits (Fig. 2g; mean diameter ± s.e.m. no laser vs. laser: $1.15 \pm 0.09°$ vs. $1.34 \pm 0.12°$, $T$-test $p < 0.01$; Wilcoxon signed rank test $p < 0.05$).

Additional analysis further demonstrated that increased sRF size after feedback inactivation could not arise by chance, due to

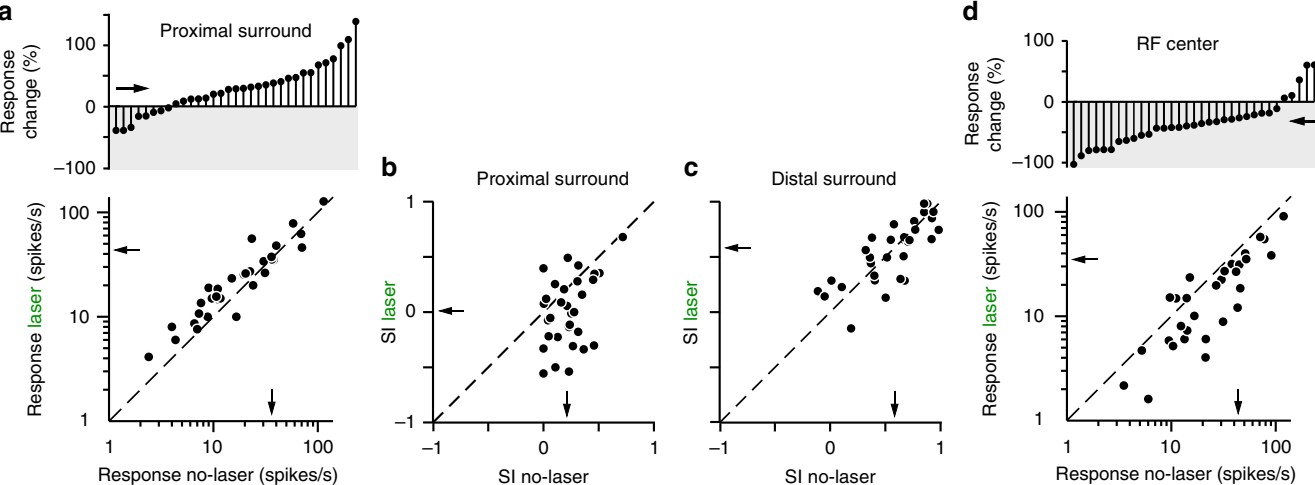

**Fig. 3** V2 feedback controls response amplitude in the sRF and proximal surround. **a**, **b** Changes in proximal surround-suppression with V2 feedback inactivated. **a** Bottom: response with and without laser for stimuli involving the sRF and proximal surround (i.e., stimulus size corresponding to the peak of the laser-on curve). Top: Cell-by-cell percent response change caused by laser stimulation, for stimuli involving the sRF and proximal surround. Downward and upward stem: decreased and increased response, respectively. **b** Suppression Index (SI; see Methods) with and without laser for stimuli extending into the proximal surround. SI = 1 indicates maximal suppression, SI = 0 indicates no suppression, and negative SI values indicate facilitation. **c** Same as **b** but for stimuli extending into the distal surround (largest stimulus used). **d** Bottom: response with and without laser for stimuli matched in size to the sRF diameter (i.e., the stimulus diameter at the peak of the empirically measured, spatial-summation curve in the no-laser condition). Top: Cell-by-cell percent response change caused by laser stimulation for stimuli matched to the sRF diameter. Arrows: means. One data point with very high firing rate in the scatterplots of panels **a**–**d** was removed for visualization purpose, but it was included in the analysis. Dashed line in **a**–**d**: unity line

noise in the data (see Supplementary Note 3 and Supplementary Fig. 4).

As feedback connections have been implicated in surround suppression, we asked whether inactivating feedback also affects the size of the RF surround. We found that whether derived from the empirical summation data or model fits to these data, the size of the surround field (see Methods for definition) was not affected by feedback inactivation either across the population (T-test $p$ = 0.33), or in individual layers (T-test $p$ > 0.27 for all layers) (see Supplementary Note 4). Because feedback connections from areas V3 and MT, which are spatially more extensive than feedback from V2[34], were unperturbed in our study, a plausible explanation for this result is that feedback connections from these areas still provide large surround fields to V1 cells when V2 feedback is inactivated.

**V2 feedback affects sRF and surround response amplitude.** Stimuli extending into the proximal surround (i.e., the surround region closest to the sRF, here defined as the stimulus diameter at the peak of the laser-on size tuning curve, e.g., Fig. 2a left panel), evoked larger neuronal responses (mean ± s.e.m. no-laser vs. laser: 36.4 ± 12.3 vs. 43.5 ± 17.2 spikes/s; mean increase 29.2 ± 7.14%, T-test $p$ < 0.001; Fig. 3a), and, therefore, less surround suppression (or even facilitation) with feedback inactivated when compared with intact feedback. Thus, not only the peak of the size tuning curve shifted towards larger stimulus sizes after feedback inactivation, but the response amplitude at this peak was also increased compared to the response with feedback intact. Note that our definition of proximal surround does not enforce this result. For example, in some cells (e.g., the one shown in Fig. 2c) the response was smaller in the laser-on condition compared to the control condition. Laser stimulation reduced the suppression index (SI; see Methods) for stimuli covering the sRF and proximal surround, measured relative to the peak response in the no-laser condition (SI no-laser vs. laser: 0.21 ± 0.03 vs. 0.006

± 0.0567, T-test $p$ < 0.01; Fig. 3b). In contrast, the responses (no-laser vs. laser: 20.9 ± 8.71 vs. 19.79 ± 7.69 spikes/s; mean spike-rate decrease 7.10 ± 13.4%, T-test $p$ = 0.92) and SI (no-laser vs. laser: 0.58 ± 0.05 vs. 0.58 ± 0.05; T-test $p$ = 0.945; Fig. 3c) evoked by stimuli extending into the more distal surround were unchanged by feedback inactivation. V2 feedback inactivation is, indeed, expected to affect most strongly proximal surround suppression, and to not abolish the most distal surround suppression. This is because feedback connections from V2 do not extend into the most distal surround regions of V1 neurons, unlike feedback connections from areas V3 and MT[34], which were unperturbed in this study. Thus, the fact that the strength of surround suppression was mostly unaffected at the largest stimulus diameters is consistent with the anatomical extent of feedback connections to V1 arising from different extrastriate areas[34].

For most (e.g., Figure 2a left panel), but not all (e.g., Figure 2a right panel) neurons, inactivating feedback also changed the neuron's response to small stimuli, the size of the neuron's sRF or smaller. We quantified these effects across the neuronal population. Consistent with previous studies of V2 inactivation[27,29], we found that across the population of cells, stimuli matched in size to the neurons' sRF diameter (i.e., the stimulus diameter at the spatial summation peak in the no-laser condition) on average evoked lower responses in the laser condition (35.1 ± 15.3 spikes/s) compared to the no-laser condition (43.8 ± 14.1; mean reduction 32.0 ± 6.03%, T-test $p$ < $10^{-5}$; Fig. 3d). Therefore, feedback inactivation reduced the amplitude of V1 neuron responses to stimuli inside the sRF. Although in some cells feedback inactivation increased neural responses to the smallest stimuli that evoked no response in the no-laser condition (e.g. Figure 2a left panel), this increase was not significant across the population (average spike-rate difference between laser and no-laser conditions 1.28 ± 0.67 spikes/s, T-test $p$ = 0.39). We also found a moderate, but statistically insignificant, relationship between response reduction to stimuli matched

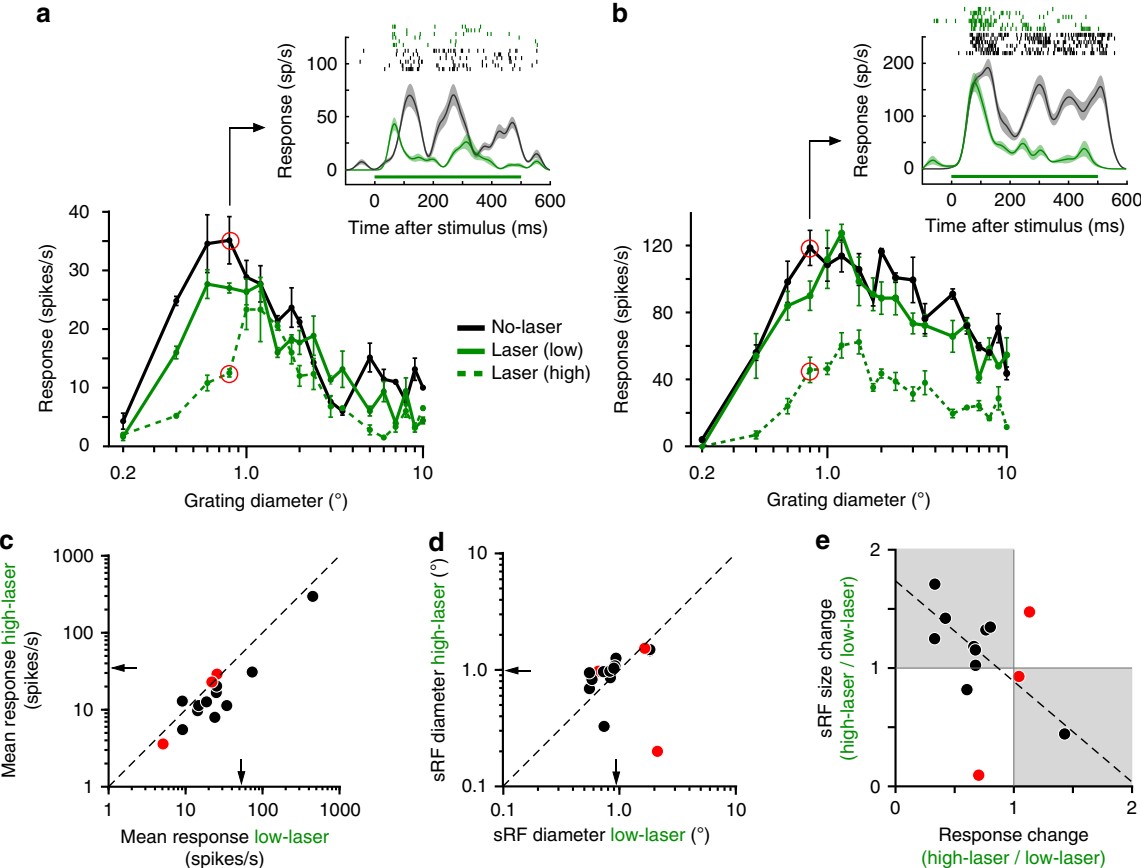

**Fig. 4** Feedback controls the amplitude of V1 responses. **a**, **b** Spatial-summation curves for two example V1 cells measured without laser (black) and with (green) laser stimulation at two different intensities (solid green: 9 mW/mm²; dashed green 43 mW/mm²). Other conventions are as in Fig. 2a. **c** Response amplitude (here defined as mean response over the entire spatial summation curve) for each cell at low and high laser intensity ($n = 14$). **d** sRF diameter for each cell at low and high laser intensity. Black and red dots in **c**–**e** indicate cells showing significant (T-test, $p < 0.05$) and non-significant change in response amplitude, respectively. Arrows in **c**–**d**: means. Dashed line in **c**–**d**: unity line. **e** sRF size change (ratio of diameter at low to high laser) vs. response change (ratio of response amplitude at low to high laser). Shaded area indicates cells for which sRF size increased and response amplitude decreased with increasing laser intensity (and vice versa). Dashed line: regression line

to the sRF diameter and change in sRF diameter when feedback was inactivated ($r = -0.31$, $p = 0.11$, Pearson's correlation), as well as between change in sRF diameter and release from suppression in the proximal surround ($r = 0.32$, $p = 0.08$).

Prolonged light pulses directed on ArchT-expressing axon terminals have been shown to facilitate synaptic transmission, while ArchT is consistently suppressive for pulse widths of ≤200 ms[35]. Thus, we also performed the analysis described above focusing only on the first 200 ms of the response. The results of the original and shorter time-scale analyses were qualitatively and quantitatively similar (see Supplementary Note 5), thus indicating that the observed results were caused by feedback inactivation.

**V2 feedback affects overall response amplitude**. The analysis above revealed that in addition to increasing sRF size, feedback inactivation also affected neuronal response amplitude. For most cells, responses to stimuli in the sRF were reduced. However, responses to stimuli extending into the surround were increased in some cells (Fig. 2a), but decreased in other cells (Fig. 2f inset). We asked whether different levels of laser intensity had different impact on V1 neurons' response amplitude.

Figure 4a, b shows two example cells in which sRF size progressively increased and response amplitude progressively decreased with increasing laser intensity. However, the cell in

Fig. 4b showed greater and overall response reduction, while for the cell in Fig. 4a response reduction was more pronounced at smaller stimulus diameters. Across the population of cells ($n = 33$) we found that 36% of neurons showed response reduction across the entire spatial summation curve, and these were the neurons in the population that showed strongest surround suppression in the no-laser condition (SI: $0.78 \pm 0.03.1\%$ vs. $0.49 \pm 0.07\%$, T-test $p < 0.05$).

We quantified how sRF diameter and mean response amplitude varied with laser intensity. This analysis is based on a population of 14 cells for which at least two laser intensities (within the range selected on the basis of the control experiments described in Supplementary Figs. 2-3) induced significant changes in the spatial summation curve (ANOVA $p < 0.05$); for each of these cells the analysis was performed at the lowest (range: 3–31 mW/mm²) and highest (range: 18–43 mW/mm²) intensity.

Compared to lower laser intensity, at higher laser intensity 11/14 cells showed a significant reduction in mean response amplitude (T-test $p < 0.05$; Fig. 4c) and 10/14 cells showed increased sRF diameter (Fig. 4d). Furthermore, most cells (10/14) showed both, reduced response amplitude and increased sRF size with increasing laser intensity (Fig. 4e). For the cells that showed a statistically significant response change at higher laser intensity ($n = 11$; black dots in Fig. 4e), there was a significant negative correlation between response change and sRF size change ($r = -0.77$, Pearson's correlation, $p < 0.01$).

These results indicate that the magnitude of the feedback effects on sRF size and response amplitude depend on the level of feedback inactivation. Stronger reduction in feedback activity leads to both progressively greater increase in sRF size and progressively greater decrease in response amplitude.

**Mechanisms underlying the effects of feedback inactivation.** We fitted models with overlapping but distinct Gaussian mechanisms interacting either divisively (ROG model) or subtractively (DOG model)[14,15] to the spatial summation data presented in Fig. 2 in the laser and no-laser conditions, and compared how well each model fitted the data (see Methods). For the majority of the cells (79%), the ROG model provided a better fit to the data (mean $R^2 \pm$ s.e.m. for cells that were best fit by the ROG model $0.67 \pm 0.04$ vs. $0.37 \pm 0.10$ for the DOG model fits to the same cells). For the reminder of cells (21%), both models provided similar good fits to the data. This result

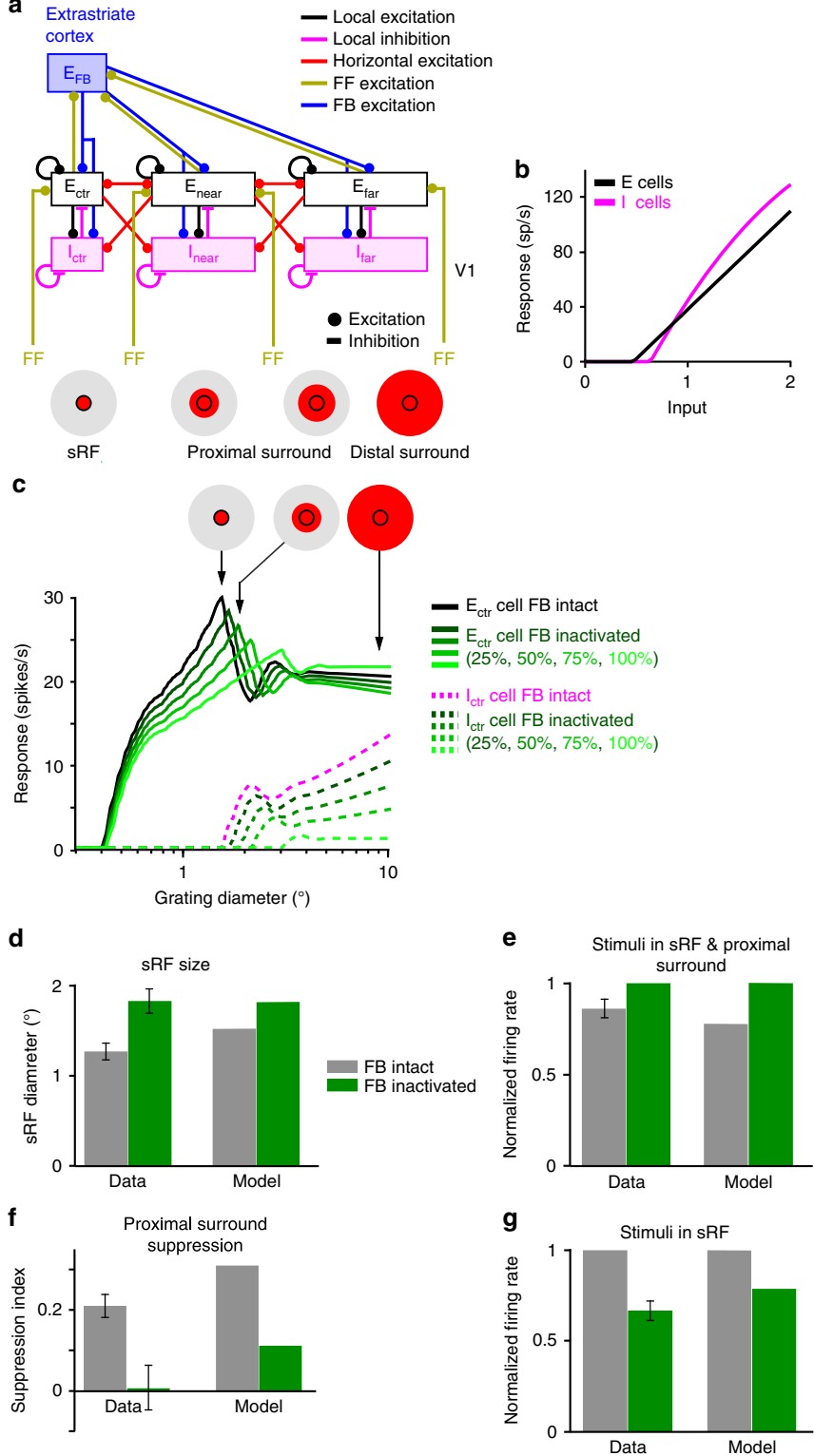

is consistent with the idea that the surround affects neural responses via divisive normalization mechanisms[14].

We next determined which model parameters were mostly affected by feedback inactivation. For each cell, we selected the model that provided the best fit to its size tuning measurements averaged over no-laser and laser conditions, and then allowed one parameter at a time to vary with feedback inactivation, while holding the remainder of the model parameters fixed to the no-laser condition values. As none of the single parameter models could account for the full range of the effects seen in the inactivation data (see Supplementary Note 6 and Supplementary Fig. 5a), we next allowed two parameters at a time to vary with feedback inactivation. The model in which both the spatial extent and gain of the center excitatory mechanism were allowed to vary best accounted for the inactivation results of 30% of cells in the population, followed by a model in which the spatial extent of both the excitatory center and inhibitory surround mechanisms were varied, which, instead, provided best fits for 21% of the cells (see Supplementary Note 6 and Supplementary Fig. 5b-c). However, none of the two-parameter models provided best fit for the majority of the cells. Moreover, when comparing the different models based on the coefficient of determination ($R^2$) distributions, rather than fraction of cells best fit by each model, we found that the different models performed similarly (see Supplementary Note 6 and Supplementary Fig. 5d). Thus, the phenomenological models did not allow us to discern between potential mechanisms by which V2 feedback affects neural responses in V1.

To gain better insights into the circuit mechanisms underlying changes in sRF size and response amplitude induced by feedback inactivation, we used the 1D recurrent neural network model of Schwabe et al.[36], which accounts for surround suppression in V1 using intra-V1 horizontal and local recurrent connections, feedback connections from a single extrastriate area, and a single population of inhibitory (I) neurons (Fig. 5a; see Methods). In this model, I neurons have higher threshold and gain than excitatory (E) neurons (Fig. 5b) and, consistent with recent findings[37], are more strongly driven by horizontal connections than the E cells whose output they control. As a result, I cells generate suppression under sufficiently high levels of excitation, but are inactive for low levels of excitation. Therefore, the local network in the model becomes more dominated by inhibition with increasing excitatory drive. For weak excitatory inputs (e.g., small visual stimuli in the sRF), I neurons are silent, but for strong inputs (e.g., large stimuli encompassing the sRF and surround), they become active (Fig. 5c dashed pink curve) and

suppress the E neurons' response (Fig. 5c black curve). Therefore, the model I neurons behave similarly to somatostatin neurons in mouse visual cortex[37], beginning to respond at larger stimulus sizes than E neurons, and increasing their response with increasing stimulus size, thus causing surround suppression.

This model has been previously shown to account for the increase in sRF size seen in empirical spatial summation measurements at low stimulus contrast[38]. We found that a similar mechanism in this model also accounts for the increase in sRF size when feedback is inactivated. Specifically, in the model, moderate reduction of feedback excitation to the V1 network weakens the response of I neurons (Fig. 5c dashed green curves), allowing E neurons to summate excitatory signals over larger visual field regions (i.e., to increase their sRF size; Fig. 5c solid green curves) until the I neurons' threshold is reached leading to suppression of E neurons (Fig. 5c green curves). Further reducing feedback excitation, as achieved by progressively increasing laser intensity, leads to both progressive increase in sRF size and progressive decrease in response amplitude (Fig. 5c solid green curves). This is consistent with the behavior of most cells in Fig. 4e (data points in the shaded squares), for which we indeed found a significant negative correlation between sRF size change and response change when laser intensity was increased. The data-model comparison shown in Fig. 5d–g demonstrates that a single mechanism in the network model can qualitatively account for the main effects of feedback inactivation, i.e. increased sRF size (Fig. 5d), increased responses to stimuli extending into the proximal surround (Fig. 5e), therefore decreased proximal surround suppression (Fig. 5f), and decreased responses to stimuli in the sRF (Fig. 5g). A 50% reduction in feedback activity in the model (Fig. 5d–g) produced a 20% increase in sRF size (vs. 44% in the data) and a 70% reduction in proximal surround suppression (vs. 95% in the data). Reducing feedback activity by 75% in the model (not shown), instead led to a 40% increase in sRF size, and 60% reduction in proximal surround suppression.

The network model could not easily reproduce the overall strong reduction in response amplitude of the entire summation curve, as seen in 36% of cells, particularly at higher laser intensity (e.g., Figure 4b), perhaps because it relies on a single inhibitory neuron type. Moreover, V1 receives feedback connections from multiple extrastriate areas, whose spatial extent increases with the area's hierarchical distance from V1[34]. As the model incorporates feedback connections at a single spatial scale, it cannot optimally reproduce the differential effects on proximal vs. distal surround suppression of removing feedback from a single area, while

**Fig. 5** Effects on spatial summation of inactivating feedback connections in a recurrent network model of V1. **a** The model architecture. Connection types are color coded according to legend. Pink and black boxes: population of layer 2/3 inhibitory (I) and excitatory (E) cells, respectively, labeled according to the position of their RFs relative to that of the cells in the center recorded V1 column; accordingly, $E_{ctr}/I_{ctr}$ are the cells in the center column, and $E_{near,far}/I_{near,far}$ those in the near and far surround, respectively. The proximal surround, as defined in this study, encompasses the near surround (which is coextensive with the spatial spread of V1 horizontal connections) and the more proximal region of the far surround, while the distal surround encompasses the more distal region of the far surround (coextensive with the full extent of feedback connections). FF: excitatory feedforward afferents from other V1 layers to layers 2/3; $E_{FB}$: excitatory feedback connections from a single extrastriate area to V1. Icons at the bottom and in panel (**c**); RF and surround components, with red areas indicating regions activated by a stimulus of increasing diameter. **b** Firing rate of the local E and I cells in the model, plotted against the input current. **c** Size tuning curves of the model $E_{ctr}$ and $I_{ctr}$ neurons with intact feedback and with different levels of feedback inactivation, as per legend. **d–g** Data-model comparison. Model results were computed by multiplying feedback weights by 0.5 (50% reduction in feedback activity). **d** Comparison of sRF size in the data (left) and in the model (right). **e** Normalized spike-rates measured at the peak of the size tuning curves with and without feedback inactivation, in the data (left) and in the model (right). Both the data and model responses were normalized to the response at the peak of the size tuning curve with feedback inactivated. **f** Suppression Index (SI) in the data (left) and model (right) for stimuli extending into the proximal surround, measured as described for Fig. 3b. **g** Normalized spike-rates with and without feedback inactivation measured at stimulus sizes corresponding to the peak of the size tuning curve with intact feedback. Both the data and model responses were normalized to. the response at the peak of the size tuning curve measured with intact feedback

leaving intact more extensive feedback from other areas. Specifically, far surround suppression in the model was weaker than in the data. Thus, future refinements of this model will have to incorporate feedback at multiple spatial scales and multiple inhibitory neuron types.

## Discussion

Our study elucidates how feedback affects neural responses in the primate early visual cortex. Reducing V2 feedback activity increased sRF size, decreased V1 cell's responses to stimuli confined to their sRF, and increased their responses to stimuli extending into the proximal surround, thus weakening surround suppression. The magnitude of these effects depended on the degree of feedback inactivation, so that stronger reduction of V2 feedback activity led to greater increase in sRF size and progressive decrease in response amplitude. Therefore, our results indicate that feedback from V2 controls sRF size, proximal surround suppression and response amplitude in V1.

Our study is the first to demonstrate that feedback is part of the network that regulates the sRF size of V1 neurons. None of the previous studies reported systematic effects of inactivating extrastriate cortex on V1 cells' sRF size[23–29]. For most of these previous studies, this is because sRF size was not measured after inactivation of higher cortical areas[23,24,26,27,29]. In two prior studies[25,28], however, spatial summation measurements similar to those performed in our study were made before and after inactivation of higher visual cortex. It is unclear why no systematic effects of inactivating extrastriate cortex on sRF size were observed in these two studies, but differences with our study that could have led to the different results include the specific cortical areas that were inactivated (macaque V2 and V3[25], or cat postero-temporal visual cortex[28], likely homologue of macaque inferotemporal cortex), inactivation methods (cooling of entire cortical area/s), and data analysis. Compared to previous studies, which silenced an entire cortical area, therefore also affecting activity in downstream cortical or subcortical areas, the strength of our approach is the selective and titrated manipulation of feedback neuron activity. This may have allowed us to reveal nuanced effects caused selectively by direct feedback to V1, which could have been missed with coarser cooling methods.

Consistent with our findings, most previous studies in anesthetized animals have reported that inactivating extrastriate cortex leads to reduced responses to stimuli inside the RF of V1 cells[23,24,27–29]. In contrast, cooling areas V2 and V3 simultaneously in awake primates produced variable effects on the magnitude of V1 RF responses, including increases and decreases[25]; this variability may have been caused by fixational eye movements, to which the small RFs of V1 neurons are particularly sensitive.

There has been a lack of consensus over which circuits generate surround suppression in V1, in particular whether this is generated subcortically and relayed to V1 via geniculocortical connections, or intracortically by V1 horizontal connections and/or feedback connections from extrastriate cortex. Current experimental evidence suggests that all these connection types, in fact, contribute to surround suppression in V1[5]. On the one hand, suppression in V1 caused by large stimuli can occur as fast as visual responses to RF stimulation[39,40], and first emerges in V1 geniculocortical input layer 4C[41]; moreover, this early suppression is untuned for stimulus orientation[39,41]. These findings suggest that the earliest untuned suppression in V1 is inherited from the lateral geniculate nucleus, where neurons also show untuned surround suppression[42,43]. On the other hand, two recent optogenetic studies in mouse have provided direct

evidence for a contribution of intra-V1 horizontal connections to surround suppression in V1[37,44].

A role for feedback in surround suppression was suggested on the basis of evidence that feedback, but not monosynaptic horizontal, connections encompass the full spatial extent of the sRF and surround of V1 neurons[34], and conduct signals 10 times faster than horizontal axons[45]. Thus, the slower conduction velocity and limited spatial extent of horizontal connections would seem inadequate to mediate fast suppression[46] arising from the more distal regions of the surround of V1 neurons[6]. However, previous inactivation studies have provided contrasting results regarding the role of feedback in surround suppression. Some studies observed weak reduction in surround suppression after cooling primate area MT[23] or V2 and V3 together[25], or cat postero-temporal visual cortex[24]. Other studies, instead, found general reduction in response amplitude, but no change in surround suppression after pharmacologically silencing primate V2[27], cooling cat postero-temporal visual cortex[28] or optogenetically silencing mouse cingulate cortex[26]. In our study, feedback inactivation caused both reduced surround suppression and changes in response amplitude, with reduced response amplitude most often observed after stronger feedback inactivation. Therefore, our results support the involvement of feedback in both surround suppression and response amplitude. The discrepancy between studies on the effects of feedback inactivation on surround suppression could be attributed to several differences, including levels and spatial extent of feedback inactivation, the specific cortical area inactivated (two of these studies inactivated higher level cortical areas), and methods of quantifying surround suppression that did not take into account the spatial extent of the specific feedback system that was inactivated.

Inactivating V2 feedback reduced suppression predominantly in the proximal surround, and did not abolish distal surround suppression. This was predicted on the basis of the known visuotopic extent of V2 feedback connections. The latter are less extensive than feedback connections arising from areas V3 and MT, which, instead, encompass the full extent of the distal surround[34].

To gain insights into the mechanisms underlying the impact of feedback on V1 neuron responses, we fitted the data with phenomenological models previously used to describe the effects of contrast on sRF size[14,38], as well as the effects of inactivating areas V2 and V3 on surround suppression in V1[47]. In these models, the RF and surround have Gaussian sensitivity profiles, with the RF described as an excitatory Gaussian and the surround as an inhibitory Gaussian, the two interacting either subtractively or divisively. Sceniak et al.[38] found that at low stimulus contrast, sRF size is larger and response amplitude is lower than at high contrast, and suggested this results from an increase in the spatial extent of the center Gaussian mechanism. Cavanaugh et al.[14], instead, demonstrated that contrast-dependent changes in sRF size and response amplitude could be explained by changes in the gain of both the center and surround Gaussian mechanisms. Our modeling results differ from these previous reports, because although the effects of contrast on sRF size and response amplitude resemble some of the effects of feedback inactivation, particularly those we have observed at higher laser intensity, they nevertheless represent only a subset of the full range of feedback inactivation effects. Thus, models in which feedback inactivation modifies only the spatial extent of the center Gaussian[38], or only the gain of both the center and surround Gaussians[14] could capture the increase in sRF size and response reduction, but failed to capture the simultaneous response decrease to stimuli in the sRF and response increase to stimuli extending into the proximal surround.

The modeling work of Nassi et al.[47] showed that changes in the spatial extent of inhibition best accounted for changes in V1 spatial summation after simultaneous cooling of macaque areas V2 and V3. In agreement with this previous study, we found that such a model could capture the changes in neural responses for stimuli extending into the surround, i.e., the reduction in surround suppression found in both our and these authors' study. However, in contrast to Nassi et al., we found that feedback inactivation also caused an increase in sRF size, and this could not be accounted for by a model in which feedback only affects the spatial extent of surround inhibition. Instead, we found that a model involving changes in the spatial extent and gain of the excitatory mechanism provided a better account for the range of feedback inactivation effects. However, the performance of the different phenomenological models was similar, and thus did not allow us to draw firm conclusions about potential mechanisms by which V2 feedback affects neural responses in V1.

A simple network model in which spatial summation results from the interaction of feedforward, V1 horizontal and inter-areal feedback connections with local recurrent networks, provided greater insights into the network mechanisms that may underlie these effects of feedback inactivation. In this model, changes in sRF size and response amplitude after feedback inactivation were explained by a single mechanism, asymmetric inhibition, which leads to an altered balance of excitation and inhibition when excitatory feedback inputs to E and I neurons are reduced. This model was in good qualitative agreement with the effects of feedback inactivation observed in the data, namely increased sRF size, decreased responses to stimuli in the sRF, increased responses to stimuli extending into the proximal surround (Fig. 5e), and therefore reduced proximal surround suppression. While in our model asymmetric inhibition is implemented using high-threshold/gain somatostatin-like inhibitory neurons, in principle other models with asymmetric inhibition should be able to account for feedback inactivation effects on sRF size and response amplitude. For example, in the model of Rubin et al.[48], asymmetric inhibitory/excitatory responses are implemented using a mechanism based on a supralinear input/output function of cortical neurons (which causes the gain of the input/output function to increase with increasing postsynaptic activity) and an inhibition-stabilized network (in which strong recurrent excitation is stabilized by strong recurrent inhibition). It will be interesting to see if this model can account for the variety of response changes induced by feedback inactivation.

Finally, it is important to point out that several forms of top-down influences in sensory processing have been shown to affect neuronal responses in the same way as we have shown here for feedback from V2. For example, spatial attention increases the response of neurons at attended locations[2,7], modulates surround suppression[8,49] and, at least in parafoveal V1, modulates RF size[9]. Our results suggest that these effects can all be mediated by top-down modulations of feedback to early visual areas.

Our study shows that V2 feedback controls the sRF size and response amplitude of V1 neurons and contributes to surround suppression in V1. Modulation of sRF size and response amplitude by feedback connections, may serve to control the spatial resolution of visual signals and perceptual sensitivity to image features.

## Methods

**Surgery and viral injections.** All procedures conformed to the guidelines of the University of Utah Institutional Animal Care and Use Committee. Each of three adult marmoset monkeys (*Callithrix jacchus*) received 2-3 injections in dorsal area V2 of a 1:1 viral mixture of AAV9.CaMKII.Cre ($3.7 \times 10^{13}$ particles/ml) and AAV9. Flex.CAG.ArchT-GFP ($9.8 \times 10^{12}$ particles/ml; Penn Vector Core, University of Pennsylvania, PA). Injections were targeted and confined to V2 using as guidance the location of the V1/V2 border identified in vivo using intrinsic signal optical

imaging. Surgical procedures were as previously described[50]. Briefly, animals were pre-anesthetized with ketamine (25–30 mg per kg, i.m.) and xylazine (1 mg per kg, i.m.), intubated, artificially ventilated with $N_2O$ and $O_2$ (70:30), and the head was stereotaxically positioned. Anesthesia was maintained with isoflurane (1-2%), and end-tidal $CO_2$, blood oxygenation level, electrocardiogram, and body temperature were monitored continuously. The scalp was opened and the skull was thinned using a dental drill over areas V1/V2, covered with agar and a coverslip, which was glued to the skull. On completion of surgery, isofluorane was turned off, anesthesia maintained with sufentanil citrate (8–13 µg per kg per hr, i.v.), and paralysis was induced with repeated 30–60 min intravenous boluses of rocuronium bromide (0.6 mg per kg per hr) to stabilize the eyes. The pupils were dilated with a topical short-acting mydriatic agent (tropicamide), the corneas protected with gas-permeable contact lenses, the eyes were refracted, and optical imaging was started. Once the V1/V2 border was functionally identified, the glass coverslip was removed, small craniotomies and durotomies were performed over V2, and the viral mixture slowly pressure-injected (240 nl per site at 500 µm and again at 1200 µm depth, using glass pipettes of 40–50 µm tip diameter, 15 min per 240 nl). The thinned skull was reinforced with dental cement, the skin sutured and the animal recovered.

**Optical Imaging.** Acquisition of intrinsic signals was performed using the Imager 3001 (Optical Imaging Ltd, Israel) under red light illumination (630 nm). Imaging for orientation and retinotopy allows identification of the V1/V2 border (Fig. 1a, b). Orientation maps were obtained using full-field, high-contrast (100%), pseudo-randomized achromatic drifting square-wave gratings of 8 orientations at 0.5–2.0 cycles per degree spatial frequency and 2.85 cycles per sec temporal frequency, moving back and forth, orthogonal to the grating orientation. Responses to same orientations were averaged across trials, baseline subtracted, and difference images obtained by subtracting the response to two orthogonal oriented pairs (e.g., Fig. 1b middle panel). Retinotopic maps were obtained by subtracting responses to monocularly presented oriented gratings occupying complementary adjacent strips of visual space, i.e., masked by 0.5–1° strips of gray repeating every 1–2°, with the masks reversing in position in alternate trials (Fig. 1b right panel)[51]. In each case, reference images of the surface vasculature were taken under 546 nm illumination (green light, Fig. 1b left panel), and later used as reference to position pipettes for viral vector injection.

**Electrophysiological recordings and visual stimulation.** Following 62–68 days after the viral vector injection, animals were anesthetized and paralyzed by continuous infusion of sufentanil citrate (6–13 µg/kg/h) and vecuronium bromide (0.3 mg/kg/h), respectively, and vital signs were continuously monitored, as described above. The pupils were dilated with topical atropine, protected with lenses and refracted. GFP-expressing V2 injection sites and V2 feedback axonal fields in V1 were identified with GFP goggles (Fig. 1e top panel), and small craniotomies were made over V1. Extra-cellular recordings were made in V1 with 24-channel linear multielectrode arrays (V-Probe, Plexon, Dallas, TX; 100 µm contact spacing, 20 µm contact diameter) coated with DiI (Molecular Probes, Eugene, OR) to assist with post-mortem reconstruction of the electrode penetrations (e.g., Fig. 1e bottom panel), and lowered normal to the cortical surface (using triangulation methods) to a 2–2.2 mm depth over 60–90 min. A 128-channel system (Cerebus, Blackrock Microsystems, Salt Lake City, UT) was used for signal amplification and digitization (30 kHz). Continuous voltage traces were band-pass filtered (0.5–14.25 kHz), and spikes were detected as spatiotemporal waveforms using the double-threshold flood fill algorithm[52] (thresholds 2 and 4 × noise S.D.). This procedure was adopted because the apical dendrites of pyramidal cells run parallel to the probe shank and may spread the same waveforms across multiple channels. A masked EM algorithm[53] was used for clustering, and manual refinement of the clusters was performed with the Klustasuite[52].

After manually locating the recorded RFs, their aggregate minimum response field was quantitatively determined using a sparse noise stimulus (500 ms, 0.0625–0.25 deg² square, luminance decrement, 5–15 trials; Supplementary Fig. 1b) and all subsequent stimuli were centered on this field. Orientation, eye, spatial and temporal frequency preferences for the cells in the recorded V1 column were determined using 1° diameter, 100% contrast drifting sinusoidal gratings monocularly presented on an unmodulated gray background of 45 cd m⁻² mean luminance. We then performed spatial summation measurements using circular patches of 100% contrast drifting sinusoidal gratings of increasing diameter centered over the columnar aggregate minimum response field. The patch diameter ranged from 0.2–0.6° to 10–18° (depending on animal) and different patch sizes were presented in random order within each block of trials. All size-tuning experiments were performed using gratings of spatial and temporal frequencies and orientation that strongly drove most cells in the column. It was not typically challenging to find spatial and temporal frequency values to which all cells in the column responded vigorously. When the penetration was perfectly vertical, orientation preference was also similar for all cells in the column. Slight deviations from vertical, however, even for RFs perfectly aligned in space, could cause orientation to shift slightly across the column, due to the narrow orientation tuning of many V1 cells[54]. In this case, the size tuning experiment was run using two different orientations. Importantly, although deviations from optimal stimulus parameters can increase the neurons' summation area[55], these deviations are not expected to cause differences between neuronal responses recorded with and

without laser stimulation. To monitor eye movements, the RFs were remapped by hand approximately every 10 min, and stimuli were re-centered in the RF when necessary. Stimuli were presented for 500 ms with 750 ms inter-stimulus interval. Stimuli were programmed with Matlab (Mathworks, Natick, MA) and presented on a linearized CRT monitor (Sony GDM-C520, 600 × 800 pixels, 100 Hz, 57 cm viewing distance) and their timing was controlled with the ViSaGe system (Cambridge Research Systems, Cambridge, UK). Data analysis was performed using custom scripts written in Matlab and Python[56,57].

**Laser stimulation**. A 532 nm laser (Laserwave, Beijing, China) beam was coupled to a 400 μm diameter (NA = 0.15) optical fiber, then expanded and collimated to a 2.8 mm spot. Reported irradiances refer to the light power exiting the collimator divided by the area of the collimator. Because the beam was collimated, the illumination spot size depended very little on the distance of the fiber from the brain. Laser timing was controlled at submillisecond precision, using custom made programs running on real-time Linux. Light was shone on the surface of V1 through thinned skull in the regions of GFP expression, and V2 was shielded from light. Laser onset was simultaneous with stimulus onset and photostimulation continued throughout stimulus presentation (500 ms). The animal's eyes were shielded from the laser light.

**Neuronal sample selection**. We analyzed 67 visually responsive (defined as max response at least 2 SD > baseline) and stimulus modulated (one-way ANOVA, $p < 0.05$) units. Approximately 61% (41/67) of these visually driven single-units were modulated by one or more laser stimulation intensities (two-way ANOVA, either laser or stimulus diameter x laser interaction, $p < 0.05$, or at least two successive data points different in the same direction, $p < 0.05$). We were not able to determine sRF size for eight cells, thus these were excluded from further analysis. Therefore, a total of 33 cells were analyzed for the results reported in Figs. 2 and 3. Figure 2c, d is based on smaller populations of cells within this larger population of 33 cells (as indicated in the figure legend), and the three populations were not mutually exclusive.

For the analysis of the data presented in Fig. 2, the laser stimulation intensity producing the largest change in sRF size (but within the range of intensities selected on the basis of control experiments- see Supplementary Figs. 2-3 and Supplementary Note 2) was determined for each unit separately, and the analysis was performed at this intensity. This was motivated by expectations that the light intensity required to produce inactivation effects differs among cells due to several factors, including variation in opsin expression across neurons, distance of the cells from the light source, and intrinsic differences in sensitivity to feedback perturbation. Importantly, however, even though we selected different light intensities for different cells, the direction of the effects was not biased by our analysis, as we selected for each cell the laser intensity causing the largest change in sRF size, irrespective of whether this was an increase or decrease.

The analysis of the data in Fig. 4 is based on a population of 14 cells for which at least two laser intensities (within the range selected on the basis of the control experiments described in Supplementary Figs. 2-3) induced significant changes in the spatial summation curve (ANOVA for either laser or stimulus diameter x laser interaction $p < 0.05$). This is a subset (14/33) of the population analyzed in Figs. 2 and 3, because for the remainder of the population we either lacked two laser intensity levels, or only one laser intensity (within the range selected on the basis of control experiments) caused significant changes.

**Definition of RF and surround size**. From the size tuning curves, measured as described above, for each cell we extracted as a measure of RF size the grating's diameter eliciting maximum response, which we term the summation RF (sRF) size. Surround size was defined as the smallest grating diameter for which the neuron's response was reduced to within 5% of the response at the largest diameter. As these measures of sRF and surround size can be subject to noise, to derive more robust measures, we also fitted the size tuning data with the ratio and difference of the integral of two Gaussian functions (ROG[14] and DOG[15] models, respectively; see below for model fits). From the fitted summation curves we extracted the cells' sRF size as the smallest stimulus diameter at which the cell response reached 95% of the peak response[14].

**Statistical Model Fitting**. ROG[14] (eq. 1) and DOG[15] (eq. 2) models were fitted to the size tuning data according to the following functions

$$R = b + \frac{g_c L_c(x)}{1 + g_s L_s(x)} \tag{1}$$

$$R = b + g_c L_c(x) - g_s L_s(x) \tag{2}$$

where

$$L_c(x) = \left( \frac{2}{\sqrt{\pi}} \int_0^x e^{(-y/w_c)^2} \right)^2 \tag{3}$$

and

$$L_s(x) = \left( \frac{2}{\sqrt{\pi}} \int_0^x e^{(-y/w_s)^2} \right)^2 \tag{4}$$

Here the variable $x$ corresponds to the diameter of the stimulus, $w_c$ and $w_s$ are the spatial extents of the center excitatory and surround inhibitory Gaussian mechanisms, respectively (with the constraint that $w_c < w_s$), $L_c$ and $L_s$ are the activities of the center and surround mechanisms, respectively, and $g_c$ and $g_s$ are the gains of the center and surround mechanisms, respectively. All parameters were constrained to positive values during optimization. Model parameters were optimized by minimizing the sum of squared errors between the model predictions and the data. Initial parameter search was done by performing two successive grid optimizations. The first grid was coarse, and the second grid was finely spaced and centered on the best fitting parameters determined with the first grid search. The best fitting parameters determined with the second grid were used as initial parameters for final optimization, which was done using the active-set algorithm in Matlab. As the models have an equal number of parameters, model comparisons were performed by directly comparing coefficient of determination ($R^2$) values. $R^2$ values were estimated using linear regression.

**Laminar border identification and analysis of RF alignment**. To ensure that the array was positioned orthogonal to the cortical surface, we used as criteria the vertical alignment of the mapped RF at each contact (see Supplementary Fig. 1b), as well as the similarity in the orientation tuning curves recorded at each contact. The array was removed from cortex and repositioned, if significant RF misalignments across contacts were detected. The degree of RF misalignment was also quantified for each penetration as described in Supplementary Note 1.

The borders between the granular layer (4 C) and supra- and infragranular layers were determined by applying current source density (CSD) analysis, using the kernel CSD method[58], to the band-pass filtered (1–100 Hz) and trial averaged ($n = 400$) continuous voltage traces evoked by a brief full-field luminance increment (100 ms, every 400 ms, 1–89 cd m$^{-2}$; Supplementary Fig. 1a). As previously established[59], the earliest current sink corresponds to the granular layer, and its borders with the supra- and infra-granular layers can be determined from the reversals from current sink to current source above and below the granular layer, respectively.

**Statistical analysis**. Statistical $p$-values refer to either independent sample or one sample two-tailed T-tests. For the within layer comparisons (Fig. 2e), where the expected effect direction was known, one-tailed $t$-tests are reported. When deviations from normality were detected using QQ-plots (RF size analysis), the $T$-tests were augmented with Mann–Whitney U-test. The variances of statistically compared groups were not significantly different (Levene's test $P > 0.2$ for RF size comparisons; F-test $p > 0.17$ for response amplitude comparisons). Unless otherwise specified, for all groups, mean ± standard error (s.e.m.) of the mean is reported.

**Suppression index**. The Suppression Index (SI) in Fig. 3b, c was computed as follows: $SI_{\text{no-laser}} = (R_{\text{C-no-laser}} - R_{\text{CS-no-laser}})/R_{\text{C-no-laser}}$. $SI_{\text{laser}} = (R_{\text{C-no-laser}} - R_{\text{CS-laser}})/R_{\text{C-no-laser}}$, where $R_{\text{C-no-laser}}$ is the response to a stimulus confined to the sRF (the peak of the summation curve) in the no-laser condition, $R_{\text{CS-no-laser}}$ is the response to the stimulus covering the sRF and surround in the no-laser condition (the proximal surround only for the measurements in Fig. 3b, and the full extent of the surround for the measurements in Fig. 3c), and $R_{\text{CS-laser}}$ is the response to the stimulus covering the sRF and surround in the laser condition.

**Histology**. On completion of the recording session, the animal was perfused transcardially with 2–4% paraformaldehyde in 0.1 M phosphate buffer. The occipital pole was frozen-sectioned at 40 μm, tangentially to the cortical surface ($n = 2$ brains), or sagittally ($n = 1$). GFP label in V2 and V1 and DiI tracks were visualized under fluorescence to ascertain injection sites were confined to V2, and electrode penetrations were targeted to regions expressing GFP (Fig. 1d, e). Electrode penetrations from regions with low GFP expression were eliminated from analysis. Sections were counterstained with DAPI (Sigma-Aldrich, St. Louis, MO) to identify V1/V2 border and cortical layers (Fig. 1d top right panel).

**Network model**. The network mechanisms underlying the observed effects of feedback inactivation were investigated using the model of Schwabe et al.[36]. We used exactly the same recurrent network architecture and parameters as in the original published model, which was shown to capture several response properties of surround suppression in V1, including contrast-dependent changes in sRF size and surround suppression strength. However, as it has since been discovered that feedback axons directly target both excitatory and inhibitory neurons in primates[60], direct feedback connections to local inhibitory neurons were included in the current model (1/10 weight compared to feedback connections to excitatory neurons), as in ref. [61].

For model details we refer the reader to the original publication. Briefly, the network model represents two areas of visual cortex, V1 and an extra-striate area, each area simplified to a single cortical layer. A schematic diagram illustrating the basic network architecture is shown in Fig. 5a. Each spatial location in the model is represented by coupled local excitatory (E) and inhibitory (I) cells, which act as the basic functional module of the network that incorporates the effects of local recurrent connections. Interactions between these modules are mediated by horizontal and feedback connections. The spatial profile and conduction velocities of horizontal and feedback connections are constrained by existing anatomical and physiological data, according to which feedback connections are spatially more extensive[34] and have faster conduction velocities[45] than horizontal connections. Because we are focusing on size-tuning effects in this study, it seemed sufficient to take a very simple local network model with a single inhibitory neuron type. The stimulations were run with 30% contrast, which is equivalent to translating the contrast response functions of the model neurons along the contrast axis. This modification is justified as V1 neurons exhibit a variety of contrast preferences.

**Data availability**. The data will be made available upon reasonable request to the authors.

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

## Acknowledgements

We thank Kesi Sainsbury for technical assistance, and Dr. Valentin Dragoi for useful comments on the manuscript. This work was supported by grants from the National Institute of Health (R01 EY026812, R01 EY019743, BRAIN U01 NS099702), the National Science Foundation (IOS 1355075, EAGER 1649923), the University of Utah Research Foundation, The University of Utah Neuroscience Initiative, to A.A., a grant from Research to Prevent Blindness, Inc. to the Department of Ophthalmology, University of Utah, and a postdoctoral fellowship from the Ella and Georg Ehrnrooth Foundation to L.N.

## Author contributions

L.N., S.M., M.B., and A.A. designed project and collected electrophysiological data. L.N, S.M., and F.F. performed optical imaging and viral injections. L.N. analyzed optogenetic and electrophysiological data. S.M. analyzed optical imaging data and histological expression of GFP label. S.M. and F.F. generated histological figures. L.N. and S.M. built the optogenetic stimulation system. A.A. supervised all aspects of project. L.N., S.M., and A.A. wrote the paper. All authors discussed the results, commented on and approved the final manuscript.

## Additional information

**Conflict of interest** The authors declare no conflict of interest.

