## [Peer Review File · Nature Communications]

Reviewers' comments:

Reviewer #1 (Remarks to the Author):

I. Overall

This study makes an important manipulation of V2 to V1 projections to examine the role of feedback to the superficial layers of V1. The results, whether negative or positive would be interesting and useful to a broad community including vision and cortical circuit scientists. The main interpretation is that the feedback manipulated here is involved in creating near surround suppression and setting overall gain.

II. Main concerns

1. The three example cells (fig 2a and fig 3a) do not show much change in the fraction of surround suppression. The size of the stimulus associated with the peak response does change, but for the larger stimuli, there is no obvious change in fractional suppression. In the second and third examples, the change in peak time seems to come down to peculiarities or noise of the curves. Given that Methods states that the laser power was chosen that gave the largest change in peak size, it seems possible that the results could arise for a variety of reasons beyond the one suggested here.

For example, the asymmetric nature of the peak of the size tuning curve (drops off quickly as you move to smaller sizes but slowly as you move toward larger sizes - noting also the log scale of horizontal axis) means that if a variety of noisy signals are added to the curve, then by random chance, the tallest peak can be more likely to move to the right, giving larger RF sizes. Some analysis and possibly simulation would be needed to address this.

2. The paper is very brief, and packed with numbers and could be made easier to read. More labels on the figures would also help.

3. The paper would be strengthened by mentioning some predictions from possible circuits, or even showing some results from models of surround suppression in V1, which it appears the authors have worked on extensively.

4. There are other changes in firing rate in the diameter tuning curves in all three examples that are at least as large in relative or absolute magnitude, and apparently more statistically significant, compared to the ups and downs that relate to the change in peak location, and these occur at or below the stimulus size of the original peak. Yet, there is no mention of how the change in feedback relates to these more significant effects.

III. Specific comments.

Line 30-32, Abstract. I was unable to understand this phrase - "we have identified a set of fundamental operations as the cellular-level mechanisms of feedback-mediated top-down modulations of early sensory processing."

Line 48-51. Introduction. The sentence defines spatial summation to be the same as surround suppression, but I thought these were different and that the former (spatial summation) does not need to involve the latter.

Line 87. The paragraph starting here is difficult to read because of the dense listing of conditions and

numbers. On one hand, the figures are not described much in the text, which would make sense if the goal was to keep the text conceptual and free of details. On the other hand, values are then listed in the text that could easily be plotted in figures and stated in figure legends.

Line 91-92. Results. If I am reading this correctly, the statement that there was a significant decrease for cells that showed a decrease carries very little information. There is not much meaning of a significance test if the sub-population is preselected to have a difference in a particular direction.

Line 98. The concept of 'surround diameter' is introduced here without any clarification to the reader of what it is, or why it is important.

Figure 2. Two sub panels in this figure have neither an x axis label or y axis label (y axis says 50% change, but not change in what).

Line 101. The statement implies causality where there might not be any. The "increased RF size" is an observation that seems basically the same as the observation that "stimuli extending into the proximal surround evoked larger neuronal responses". There does not seem to be any evidence of the latter being the result of the former.

Figure 3. There are very few labels on the figure, which make the data more difficult to comprehend. For example, panels c, d and e could have some labels to tell the reader what is different, but there are none.

Figure 3f. SG, G and IG are not defined in this figure/legend.

Line 155. Discussion. The opening paragraph of the discussion covers many complex issues in just a few phrases. It would strengthen the paper to consider pitfalls of the approach and weigh alternative hypotheses. A model is mentioned, and it would be useful for there to be predictions based on some of the models for which this data is seen to be supportive.

Line 385-386. Methods. The laser power that caused the largest change in RF size was used for each cell in the analysis of figure 2. This sounds like it would bias the change in RF size to be large.

Reviewer #2 (Remarks to the Author):

The paper: "Top down feedback controls spatial summation and response gain in primate visual cortex" investigates the role of feedback from marmoset V2 to V1 in changing receptive field profiles (spatial summation) and neuronal responses (response gain). They injected the combination of AAV9.CaMKII.Cre and AAV9.Flex.CAG.ArchT-GFP into V2 to later (following expression) inhibit V2 terminals in area V1, while probing V1 response properties to drifting oriented gratings of different sizes. This is an interesting study, but I think it requires further quantitative analysis, supply of additional data/control, and clarification.

Main points:

- Is there a specific reason why the combination of Cre-Flex was injected, rather than straightforward e.g. AAVX-CaMKII.ArchT-GFP.xxx?
- What was the alignment of RF centers for different depth? Having closely aligned centers is crucial for size tuning when recording multiple cells simultaneously. Some quantification should be reported. The data shown in figure S3b are not sufficient as they only show a depth range of 400um from which

it is not possible to determine whether electrodes were normal to the surface, or what the alignment across depths was. For this a wider range >1.2mm needs to be shown.

- It is stated that optimal orientation, SF and TF stimuli were presented for the inactivation experiments. This has to be some compromise for the different cells recorded along the cortical depth as they will not all match. How was the compromise reached?
- It should be described in the methods how RF size was determined/defined.
- Was inactivation confirmed by recording from injection sites in V2?
- The way RF size was determined is not explicitly identified. From the manuscript, I assume it was simply by taking the peak of the summation curve? If so, it may not be the ideal measure as it is subject to noise. Noise reduction can be achieved by taking the sample of data points into account, i.e. by some curve fitting.
- The curve fitting should also be employed to deduce additional variable that have in the past often been used to quantify spatial summation, by e.g. using a ratio of Gaussian and/or difference of Gaussian fit. This would better allow to get an idea of the potential underlying parameters that are changed by feedback inactivation (excitation gain, inhibition gain, summation field size, inhibition field size). Model comparison would also allow insight into the type of inhibition, as a better fit of an ROG would hint at different inhibition mechanisms when compared to DOG (and vice versa).

Reviewer #3 (Remarks to the Author):

This study is a technical tour de force. Using primate cortex it incorporates the expression of ArchT to inactivate the terminals in V1 of feedback neurons from V2. The paper shows conclusive effects of the results of inactivation. The main interpretation of the results comes in the discussion.

"Depending on its level of activity, feedback from V2 controls RF size, surround suppression, and the overall gain of neuronal responses in V1. Changes in RF size can dynamically alter the visual system's spatial resolution; increasing surround suppression provides increased efficiency of coding natural images^{26,27}; increasing response gain improves sensitivity to image features."

Is the correct interpretation of this statement that with normal feedback intact the response gain of the classical receptive field (in spikes/sec/deg) is greater than when feedback is inactivated? Then there is an increase in the response at somewhat larger diameters (in the proximal surround) during inactivation. Since there is also an increase in the size during inactivation the slope of the rising phase of the area summation curve is steeper with feedback intact, hence feedback is contributing to the gain within the classical receptive field. This would be somewhat analogous to increasing the contrast? At larger sizes inactivation reduces the efficacy of the proximal surround with a resultant increase in response and reduction in the proximal suppression.

This is a compelling result. If the authors were able model the area summation incorporating various factors that the Angelucci group has already shown to be central the generation of different parts of the area summation function these results would reach a wider audience. As they are presented it takes a lot to disentangle the important aspects and fit them into the recent models of how the CRF, the proximal surround and extended surround are influenced by feedback from extrastriate cortex. However if the authors could use model fitting to their data to evaluate various changes it would add substantially to the understanding of the results.

The authors pitch their study as a effort to resolve different results from earlier studies. Some of the studies involved other species. The current paper shows there needs to be a nuanced interpretation of results, a number of factors are clearly involved. It might be the case that an analysis using models with different components that could be accomplished with the current results would allow for an

informed interpretation of the earlier studies. As the authors point out feedback from V2 might be different from feedback from V3 or MT, and this might depend on the specific functional class of neuron in V1 and its laminar location.

Specific Points.

1. About 36% of neurons that showed reduced surround suppression and/or increased RF size at low laser intensities showed overall reduced response at higher laser intensities (mean±sem 36.1±1.52 mW/mm²) (Fig.3a).

It is not clear how the neurons showing general response reduction were selected. Are the 12 cells analyzed for Fig 3 36% of the cells that showed effects at low intensities, shown in Suppl Fig. 2?

1. Pge 3 Fig 1b, Intrinsic imaging not explained to readers, only those in the know will understand and even some will not know how to interpret the retinotopy panel of Fig. 1b.

2. There are very careful controls for the effects of direct laser stimulation. This is clearly an issue that the authors felt it was important to address and they have done a thorough set of controls. There were some residual direct effects at some laser stimulus levels. In Fig. 1a there is a notable elevation what appears to be the non-visually stimulated response, seen at the smallest diameters. It would be very helpful to the interpretation to show what happens on a cell-by-cell basis to the spontaneous level for the laser intensities used for the inactivation vs the level without laser stimulation and visual stimulation.

3. Approximately 61% (40/66) of single units were significantly modulated by the laser (see Methods).

The numbers were hard to follow: In Fig 2 there were N = 25 cells that showed increase in RF size (Fig. 2 b1), N = 12 that had increase in RF size and peak response (Fig. 2 b2), and all cells (N = 33) in Fig. 2 b3. Additionally there were cells (N = 12) showing general suppression (Fig. 3). Maybe the categories are not mutually exclusive? But at first glance this adds up to 49 (25 + 12 + 12) or 45 (33 + 12). The authors would help the reader a lot by making the numbers clearer.

4. The PSTH's in Fig 2 have the response starting at the onset of the stimulus. Is this a consequence of the smoothing filter? Either change the filter or explain this, otherwise readers will think there is something amiss here. There is always a visual delay.

It would also seem to be the case that an analysis of the PSTH dynamics could give further insights into the mechanisms.

5. Analysis of classical RF diameter, the proximal surround size and surround field size (suppl results). There are no details of how these were estimated. It would enhance the paper to either give the methods (were the responses fitted with a model and parameters extracted) or give references to the methods if the method is already published.

6. A couple of minor notes relating to the controls show in the Supplementary data. In Fig.1a top row, 2nd and 3rd columns the laser condition has responses greater than the no laser condition for all diameters. Yet the summary plot Fig. 1c shows no points above the unity line. It would be helpful to say (as indicated in 5 above) how the peak diameter or proximal surround position were determined for each curve so readers could work out how to go from the curves in Fig 1a to the summary in Fig. 1c.

Also the responses in this data set are referred to as responses from "contacts". Does this mean this is multiunit data? The rates are substantially higher than for the data shown in the main figures. If so, it would be important to indicate that the data is multiunit.

Minor points:

1. "In contrast, the responses (no-laser vs. laser: 20.9 ± 8.71 vs. 19.79 ± 7.69 spikes/s; mean spike-rate increase $7.10 \pm 13.4\%$, $p=0.92$)."

Do you mean "spike-rate decrease" here?

2. In Suppl doc:

"This is also suggested by modeling the effects of feedback inactivation (Fig. 4b)."
There doesn't seem to be a Fig 4 or any modeling in the manuscript.

REPLY TO THE REVIEWERS

We thank the Reviewers for their thoughtful comments and constructive criticism. We have revised the manuscript accordingly. To facilitate the review of the revised version, the changed text in the revised manuscript is in red font. The following is our point-by-point reply to the Reviewers, with our answers in red text.

REVIEWER I

I. Overall

This study makes an important manipulation of V2 to V1 projections to examine the role of feedback to the superficial layers of V1..... and setting overall gain.

II. Main concerns

1. The three example cells (fig 2a and fig 3a) do not show much change in the fraction of surround suppression. The size of the stimulus associated with the peak response does change, but for the larger stimuli, there is no obvious change in fractional suppression”.

Answer 1: The Reviewer is correct in that there is little or no change in the strength of *distal* surround suppression. Indeed, this is a specific prediction of our previous anatomical studies (Angelucci et al. 2002) demonstrating that V2 feedback connections to V1 are spatially less extensive than feedback connections arising from V3/DM and MT. Specifically, unlike feedback from V3 and MT, V2 feedback connections do not extend to the most distal surround regions of V1 neurons. Therefore, inactivation of V2 feedback connections, while leaving V3 and MT feedback activity unperturbed, is expected to affect most strongly the suppression arising from the proximal surround, and to not abolish the most distal surround suppression. This issue was discussed in the previous version of the manuscript (lines 111-115). However, we now further emphasize this important prediction derived from the anatomy of feedback connections in both the Results (lines 173-175 of revised manuscript) and Discussion (lines 364-372).

“In the second and third examples, the change in peak time seems to come down to peculiarities or noise of the curves. Given that Methods states that the laser power was chosen that gave the largest change in peak size, it seems possible that the results could arise for a variety of reasons beyond the one suggested here.

For example, the asymmetric nature of the peak of the size tuning curve (drops off quickly as you move to smaller sizes but slowly as you move toward larger sizes - noting also the log scale of horizontal axis) means that if a variety of noisy signals are added to the curve, then by random chance, the tallest peak can be more likely to move to the right, giving larger RF sizes. Some analysis and possibly simulation would be needed to address this”.

Answer 2: We agree with the Reviewer that RF size estimates based on the peak of the size tuning curves can be susceptible to noise. A standard approach to minimize the influence of noise is to fit a function to the data. However, this second approach is subject to different kinds of errors, such as fitting errors. When preparing the previous version of the manuscript, we decided to avoid potential errors arising from the fact that any function is an approximation of the real underlying curve and makes assumptions about the underlying mechanisms. However, to address the criticism of two of the Reviewers, we feel the best approach is to report RF size estimates based on both empirical

measurements as well as model fits to these measurements (as we have also done in previous studies). Thus, in the revised version of the manuscript we have now included one additional measure of RF size based on fitting a ratio or a difference of integrals of Gaussians to the data (ROG & DOG models, respectively). Fig. 2c,d now shows the effects of feedback inactivation on V1 cells RF size, using two different ways of estimating RF size. Importantly, even when the RF sizes were estimated from the fits to the summation data, feedback inactivation was still found to increase average RF size, to a statistically significant extent (see Results lines 127-143 and Figure 2c-d; Methods lines 566-592).

Moreover, as suggested by the Reviewer, we have also performed a simulation to estimate the magnitude of RF size change expected to arise from noise, when RF size is defined as the peak of the empirically measured size tuning curve. We find that noise can account for a median increase in RF size of only 3.6% compared to the 56% increase seen in the data. We have included a detailed description and results of this simulation to the Supplementary Material (see Supplementary Results lines 79-95 and Supplementary Figure 4). We now also refer to this analysis in the main Results section (lines 144-146).

“2. The paper is very brief, and packed with numbers and could be made easier to read. More labels on the figures would also help”.

Answer 3: The paper was initially submitted to Nature and then transferred in this initial format to Nature Communication, which explains its compacted format. However, we have now extended the description of the Results (which are now subdivided into subsections) and the Discussion, transferred many numbers to the figure legends, and added labels to all figures, as suggested by the Reviewer.

“3. The paper would be strengthened by mentioning some predictions from possible circuits, or even showing some results from models of surround suppression in V1, which it appears the authors have worked on extensively”.

Answer 4: We have now added a modeling section to the Results (*Mechanisms Underlying the Effects of Feedback Inactivation*, see lines 226-299), to the Supplementary Material (*Phenomenological modeling*, lines 120-165), to the Methods (lines 637-656), and to the Discussion (lines 373-417) and two new figures reporting modeling results (Figs. 5 and Supplementary Fig. 5). We have used both phenomenological modeling (DOG and ROG models), as well as performed simulations using a network model we developed in 2006 (Schwabe et al. 2006) to investigate the network mechanisms that may underlie the effects of feedback inactivation.

The statistical models suggest that changes in RF size and the variety of effects on response amplitude we see after feedback inactivation can be best explained by a change in the spatial extent and gain of the excitatory center Gaussian mechanism.

The network model, on the other hand, provides a better intuition about possible circuit mechanisms. Because the effects of feedback inactivation on RF size and response amplitude are reminiscent of the effects of lowering stimulus contrast, we asked whether a network model of V1 (Schwabe et al., 2006) previously used to account for contrast-dependent RF size changes could also account for RF size changes with feedback inactivated. We found that a single mechanism similar to that underlying the expansion of the RF size at low stimulus contrast, in this model, also accounts for the increase in RF size and response amplitude changes when feedback is inactivated. This mechanism is based on inhibitory neurons being activated by larger stimuli relative to excitatory neurons, which in our model is implemented as inhibitory (I) neurons having higher threshold and gain than excitatory (E) neurons. While the biological plausibility of these I neurons remains to be fully demonstrated (but see Angelucci et al. Ann Rev. Neurosci. 2017 for an extensive discussion of this model and its biological basis), in principle other models that can implement asymmetric inhibition (e.g. the

supralinear ISN of Rubin et al. Neuron 2015) should also be able to account for these effects of feedback inactivation.

In this network model, moderate reduction of feedback excitation to the V1 network, weakens the response of I neurons, allowing E neurons to pool excitatory signals over larger visual field regions (i.e. to increase their RF size) until the I neurons's threshold is reached leading to suppression of E neurons. Stronger reduction of feedback activity in the model leads to even greater increases in RF size and reduction of response gain. Prompted by this modeling results, we re-analyzed the data at different laser intensities (former Fig. 3) on a cell-by-cell basis, rather than averaging RF size across the population at low and high laser. This led to the discovery that most cells indeed behave as our model cells, i.e. with increasing laser intensity they show a significant negative correlation between RF size increase and gain reduction. The new data analysis is now reported in new **Fig. 4**.

The Results section of the paper also discusses weaknesses of this model and what aspect of an extended model would likely provide an even more detailed account of the effects of inactivating feedback at different levels (lines 294-304).

“4. There are other changes in firing rate in the diameter tuning curves in all three examples that are at least as large in relative or absolute magnitude, and apparently more statistically significant, compared to the ups and downs that relate to the change in peak location, and these occur at or below the stimulus size of the original peak. Yet, there is no mention of how the change in feedback relates to these more significant effects”.

Answer 5: The Reviewer is correct in that inactivating feedback significantly changes responses at the stimulus size of the original peak. Indeed, **Figure 2d** of the previous version of the manuscript (**Fig. 3b** of the revised version) is a quantitative description of these changes, and demonstrates that across the population there is a significant reduction in response to stimuli confined to the neuron's RF when feedback is inactivated. However, in the previous version of the manuscript, we did not analyze changes that occurred at the smallest stimulus sizes, as noted by the Reviewer. We have now performed a systematic analysis for the latter, and found that there are no consistent changes across the population in response to the smallest stimuli (that do not cause the cell to spike under control conditions), when feedback is inactivated. The results of this new analysis are now reported in the revised version of the manuscript (lines 176-178, 183-188).

“III. Specific comments.

Line 30-32, Abstract. I was unable to understand this phrase - "we have identified a set of fundamental operations as the cellular-level mechanisms of feedback-mediated top-down modulations of early sensory processing."

Answer 6: This sentence now reads “we have identified the cellular mechanisms of feedback-mediated modulations of early sensory processing.” (see line 30).

“Line 48-51. Introduction. The sentence defines spatial summation to be the same as surround suppression, but I thought these were different and that the former (spatial summation) does not need to involve the latter”.

Answer 7: This definition has now been clarified (see lines 46-48).

“Line 87. The paragraph starting here is difficult to read because of the dense listing of conditions and numbers. On one hand, the figures are not described much in the text, which would make sense if the goal was to keep the text conceptual and free of details. On the other hand, values are then listed in the text that could easily be plotted in figures and stated in figure legends”.

Answer 8: Most of these numbers have now been moved to the figure legend and the description of the results simplified and clarified (lines 110-118 and Figure legend 2 lines 927-929).

“Line 91-92. Results. If I am reading this correctly, the statement that there was a significant decrease for cells that showed a decrease carries very little information. There is not much meaning of a significance test if the sub-population is preselected to have a difference in a particular direction.

Answer 9: Figure 2b of previous and current version of the manuscript shows changes in RF size for 2 preselected populations of neurons (panels b2 and b3), as well as for the whole population (panels b1 and b4); importantly the main effect is also present across the whole population, albeit of course its magnitude is smaller. The reason for selecting subpopulation of neurons is to show the magnitude of the effect for those cells that show the main effect of the study. We feel this is justified because there is no reason to believe that feedback must affect the whole of V1, independent of cell type, layer, or whether cells receive direct vs. indirect V2 feedback inputs. However, the Reviewer is correct that a significance test does not make sense for the pre-selected populations and therefore we now report the bar graphs without reporting statistical significance. We also emphasize in the text that the purpose of Fig. 2b2-2b3 is to show the magnitude of the RF size increase in the population of cells showing the effect (lines 115-118).

“Line 98. The concept of 'surround diameter' is introduced here without any clarification to the reader of what it is, or why it is important”.

Answer 10: The Methods now contain a section in which the definitions of the different receptive field and surround components and how they were measured in this study are explained (see lines 555-564). We now also introduce the concept of surround diameter when we first refer to this, at the specific passage mentioned by the Reviewer (now lines 147-150). Moreover RF size definitions are also defined in the results when first introduced (see lines 111-112, 140-141)

“Figure 2. Two sub panels in this figure have neither an x axis label or y axis label (y axis says 50% change, but not change in what)”.

Answer 11: We have now labeled all axes in Figs. 2, 3, 4 and Supplementary Fig. 3.

“Line 101. The statement implies causality where there might not be any. The "increased RF size" is an observation that seems basically the same as the observation that "stimuli extending into the proximal surround evoked larger neuronal responses". There does not seem to be any evidence of the latter being the result of the former”.

Answer 12: We did not mean to imply causality, but indeed meant to say the two are the same phenomenon. We have deleted that sentence (see line 158).

“Figure 3. There are very few labels on the figure, which make the data more difficult to comprehend. For example, panels c, d and e could have some labels to tell the reader what is different, but there are none”.

Answer 13: The explanation of how those panels differed from each other was in the figure legend. However, we have now added labels directly to all figures. The particular figure the Reviewer is referring to has changed almost completely so former panel c, d and e no longer exist.

“Figure 3f. SG, G and IG are not defined in this figure/legend”.

Answer 14: Figure legend 2 now contains layer abbreviations.

“Line 155. Discussion. The opening paragraph of the discussion covers many complex issues in just a few phrases. It would strengthen the paper to consider pitfalls of the approach and weigh alternative hypotheses. A model is mentioned, and it would be useful for there to be predictions based on some of the models for which this data is seen to be supportive”.

Answer 15: The Discussion section has been significantly expanded and the first paragraph changed. The discussion now contains a section on the mechanisms underlying feedback inactivation effects based on the modeling work (see lines 374-416). We also more extensively discuss our results in the context of previous studies.

“Line 385-386. Methods. The laser power that caused the largest change in RF size was used for each cell in the analysis of figure 2. This sounds like it would bias the change in RF size to be large”.

Answer 16: It is expected that the light power required to produce inactivation effects varies between cells for several reasons, including variation in opsin expression across neurons, distance of cells from the light source, and intrinsic differences in sensitivity to feedback perturbation (just as, for ex, cells have different sensitivity to stimulus contrast).

Had we chosen to analyze a single laser power for all cells, the reported effects could have been lost in averaging, thus we decided to choose the laser power independently for each cell. It is important to notice, however, that the *direction* of the effects was not in any way constrained by our analysis, because the laser power causing either the largest increase or decrease in RF size was selected for each cell. Due to space constraints, the rationale behind selecting laser power individually for each neuron was not very well explained in the original manuscript, but we have now included a more thorough discussion of this issue in the revised version (see lines 539-546).

REVIEWER 2

“The paper: “Top down feedback controls spatial summation and response gain in primate visual cortex” investigates the role of feedback from marmoset V2 to V1 in changing receptive field profiles (spatial summation) and neuronal responses (response gain). They injecteddata/control, and clarification.

Main points:

- Is there a specific reason why the combination of Cre-Flex was injected, rather than straightforward e.g. AAVX-CaMKII.ArchT-GFP.xxx?”

Answer 17: This viral vector mix was used because in pilot studies we discovered that it produces selective anterograde infection of neurons at the injected site in non-human primates, and virtually no retrograde infection of neurons in V1 (see **Figure 1d**). While we are not completely certain about the biological mechanism causing this behavior, we noticed that retrograde infection seems to be subject to a threshold effect, so that smaller volume injections yield less retrograde infection compared to larger viral injections. Based on this observation, we believe the Cre-Flex mixture works by reducing the likelihood of retrograde uptake of the virus, thus making retrograde label virtually non-existent. However, as this hypothesis is not tested, we refer only to the empirical observations from our pilot studies in the manuscript (see lines 70-72).

“• What was the alignment of RF centers for different depth? Having closely aligned centers is crucial for size tuning when recording multiple cells simultaneously. Some quantification should be reported. The data shown in figure S3b are not sufficient as they only show a depth range of 400um from which it is not possible to determine whether electrodes were normal to the surface, or what the alignment across depths was. For this a wider range >1.2mm needs to be shown”.

Answer 18: We took special care to position the electrode orthogonal to the cortical surface using triangulation methods. To further ensure that the receptive fields of simultaneously recorded neurons were vertically aligned during the recordings, minimum response fields (mRF) were first mapped by presenting 0.5 x 0.5° black squares for 500 ms in locations defined by a 3x3° grid centered on the hand mapped RF of the visually responsive units. The mRFs of all recorded units were then plotted on a screen, and in the cases where misalignment was noticed, the probe was retracted and reinserted. The Reviewer misinterpreted the original Supplementary Fig. 3b (currently Supplementary Fig. 1b) as this figure actually showed almost the full length of the penetration from 0.3 mm cortical depth to 1.2 mm; for illustration purposes we had split this penetration in two columns which probably led the Reviewer to think that the two columns represented 2 separate penetrations. We now show the RFs across the penetration in a single vertical column, and specify in the legend that the figure shows a single penetration (see Supplementary Fig. 1b). Moreover, we have also quantified the amount of misalignment for all our penetrations, as the euclidean distance of each mRF center from the mean of mRF centers in the same penetration. We found that the median euclidean distance from the average mRF center was $0.124^{\circ} \pm 0.029^{\circ} / 0.035^{\circ}$ (95% CI lower bound/upper bound; bootstrap). This analysis is now reported in the Supplementary Results (see lines 4-12).

“• It is stated that optimal orientation, SF and TF stimuli were presented for the inactivation experiments. This has to be some compromise for the different cells recorded along the cortical depth as they will not all match. How was the compromise reached?”

Answer 19: We chose orientation, SF and TF values that strongly drove all cells in the recorded column. This was easily achieved for SF and TF, as cells in V1 are typically broadly tuned for SF and TF, but can be trickier for orientation as orientation-tuning can be very narrow for some cells. When the penetration was perfectly vertical, orientation preference was similar for all cells in the column. Slight deviations from vertical, however, even for perfectly aligned spatial RFs, could cause orientation to shift slightly across the column. In the latter case, we ran the size tuning experiments using two different orientations. This is now all specified in the Methods section (see lines 497-507). It is worth pointing out, however, that while deviations from the optimal stimulus parameters can increase the

summation area of the recorded neurons, these deviations are not expected to cause differences between responses recorded with and without laser stimulation. Therefore, we think that slight deviations from optimal stimulus parameters would not alter the interpretation of our results.

“• It should be described in the methods how RF size was determined/defined”.

Answer 20: See Answer 10 to Reviewer 1 above.

“• Was inactivation confirmed by recording from injection sites in V2?”

Answer 21: Inactivation of V2 cells is not

the right control for our study, as this is exactly what we were trying to avoid. Our study was specifically designed to selectively inactivate feedback neurons, not the whole of V2. This is because, inactivation of V2 cells would open up the possibility that the observed results could be caused by indirect shutdown of circuits other than the V2 feedback, for example via the thalamus or other cortical areas to which V2 projects. We now emphasize this in line 60, as well as in lines 77-82, and have modified the cartoon of Fig. 1c to better convey this message.

However, to verify the general effectiveness of ArchT in silencing neurons in marmosets, we indeed confirmed that V2 neurons were silenced when directing the laser onto V2 (see Figure 1). We did not add this figure to the Supplementary Material because we do not think it is relevant to the study, for the reasons explained above.

“... The way RF size was determined is not explicitly identified. From the manuscript, I assume it was simply by taking the peak of the summation curve? If so, it may not be the ideal measure as it is subject to noise. Noise reduction can be achieved by taking the sample of data points into account, i.e. by some curve fitting.

Answer 22: RF size is now defined (see Answer 10 above). The Reviewer is correct as to how we defined RF size, and the fact that this measure of RF size is subject to noise. Following the Reviewer’s advice, as well as comments from Reviewer 1, we have fitted the data to DOG and ROG models (see Answer 2 to Reviewer 1 above for results of this analysis).

“ The curve fitting should also be employed to deduce additional variable that have in the past often been used to quantify spatial summation, by e.g. using a ratio of Gaussian and/or difference of Gaussian fit. This would better allow to get an idea of the potential underlying parameters that are changed by feedback inactivation (excitation gain, inhibition gain, summation field size, inhibition field size). Model comparison would also allow insight into the type of inhibition, as a better fit of an ROG would hint at different inhibition mechanisms when compared to DOG.

Answer 23: In the revised version of the manuscript, we have added a section to the Results dealing with phenomenological modeling (as well as network modeling) and possible mechanisms of feedback

inactivation (Results lines 227-255, Supplementary Material lines 120-165, Supplementary Fig. 5). See also Answer 4 to Reviewer 1 above.

REVIEWER 3

“This study is a technical tour de force. Using primate cortex it incorporates the expression of ArchT to inactivate the terminals in V1 of feedback neurons from V2. The paper shows conclusive effects of the results of inactivation.....gain improves sensitivity to image features.”

“Is the correct interpretation of this statement that with normal feedback intact the response gain of the classical receptive field (in spikes/sec/deg) is greater than when feedback is inactivated?”

Answer 24: Yes, this interpretation is correct.

“Then there is an increase in the response at somewhat larger diameters (in the proximal surround) during inactivation. Since there is also an increase in the size during inactivation the slope of the rising phase of the area summation curve is steeper with feedback intact, hence feedback is contributing to the gain within the classical receptive field. This would be somewhat analogous to increasing the contrast?”

Answer 25: Indeed, the effect is similar to changing stimulus contrast in that both decreasing contrast or feedback activity causes a reduction in response gain inside the RF, but also an increase in RF size (i.e. a shift of the peak of the size tuning curve to the right). However, lowering contrast typically shifts the entire size tuning curve downwards, whereas slight reduction of feedback activity mainly shifts the curve to the right, and in many instances even increases the response to stimuli in the proximal surround. This maybe a genuine difference or reflect the degree to which contrast is lowered in typical contrast experiments. Indeed, when feedback is more strongly inactivated at higher laser intensity, gain is reduced and the whole summation curve can be shifted down, just like the summation curve at low contrast. We have now made the similarity of inactivating feedback with manipulations of stimulus contrast more explicit in the Discussion section (see lines 378-390 see also 277-279).

“At larger sizes inactivation reduces the efficacy of the proximal surround with a resultant increase in response and reduction in the proximal suppression”.

Answer 26: Correct.

“This is a compelling result. If the authors were able model the area summation incorporating various factors that the Angelucci group has already shown to be central the generation of different parts of the area summation function these results would reach a wider audience. As they are presented it takes a lot to disentangle the important aspects and fit them into the recent models of how the CRF, the proximal surround and extended surround are influenced by feedback from extrastriate cortex. However if the authors could use model fitting to their data to evaluate various changes it would add substantially to the understanding of the results.

The authors pitch their study as a effort to resolve different results from earlier studies. Some of the studies involved other species. The current paper shows there needs to be a nuanced interpretation of

results, a number of factors are clearly involved. It might be the case that an analysis using models with different components that could be accomplished with the current results would allow for an informed interpretation of the earlier studies. As the authors point out feedback from V2 might be different from feedback from V3 or MT, and this might depend on the specific functional class of neuron in V1 and its laminar location.”

Answer 27: We have now performed phenomenological model fitting and network model simulations to get better insights into potential mechanisms (see Answers 4 and 23 above).

“Specific Points.

1. About 36% of neurons that showed reduced surround suppression and/or increased RF size at low laser intensities showed overall reduced response at higher laser intensities (mean±sem 36.1±1.52 mW/mm²) (Fig.3a).

It is not clear how the neurons showing general response reduction were selected. Are the 12 cells analyzed for Fig 3 36% of the cells that showed effects at low intensities, shown in Suppl Fig. 2?”

Answer 29: No, they were 36% of the cell population analyzed for Fig. 2. The criteria for selecting the cells showing general suppression were explained in the Methods section of the previous version of the manuscript (previous lines 378-393). However, former Fig.3 (now Fig. 4) has substantially changed, and the cells were selected on the basis of different criteria, namely we now select all cells for which we had two different levels of laser intensity that induced statistically significant changes in the summation curve. Regardless, we now better clarify the number of cells that were used for each analysis and how these cells were selected (see Methods lines 527-535, and 547-553).

Former Supplementary Fig. 2 (now Supplementary Fig. 3) shows the same analysis as in **Figs. 2-3**, but excluding SG cells showing effects at irradiances > 19mW/mm², so this figure includes a smaller population than that analyzed in Figs. 2-3. The rationale for excluding SG cells >19mW/mm² in the analysis of Supplementary Fig. 3 was that in cortex not expressing opsins, we observed in some SG cells non-specific effects of laser stimulation at irradiances >19mW/mm². Therefore, we wanted to make sure that the main results of our study still held when removing from analysis all SG cells that showed any effects at irradiances >19mW/mm². And indeed we found this is the case. However, in Figures 2-3 we still included SG cells that showed effects at irradiances >19mW/mm² (but <43 mW/mm²), because the rare effects we observed in control cortex at irradiances >19 mW/mm² but <43 mW/mm² were always in the opposite direction to the results obtained in ArchT-expressing cortex, namely a leftward shift of the peak of the size tuning curve (i.e. a decrease in RF size). Therefore, we felt these effects could not account for our main result. We have now better clarified the rationale for this analysis and added additional control analysis at different laser intensities (see lines 32-51 of the Supplementary Material, and Supplementary Fig. 2 b-c).

“1. Pge 3 Fig 1b, Intrinsic imaging not explained to readers, only those in the know will understand and even some will not know how to interpret the retinotopy panel of Fig. 1b.”

Answer 31: Intrinsic imaging is now better explained in the legend to Fig. 1 (lines 895-902) and the figure itself has been modified by adding insets above the orientation and retinotopy panels (Fig. 1b) showing the specific visual stimuli used to obtain the images.

“2. There are very careful controls for the effects of direct laser stimulation. This is clearly an issue that the authors felt it was important to address and they have done a thorough set of controls. There were some residual direct effects at some laser stimulus levels. In Fig. 1a there is a notable elevation what

appears to be the non-visually stimulated response, seen at the smallest diameters. It would be very helpful to the interpretation to show what happens on a cell-by-cell basis to the spontaneous level for the laser intensities used for the inactivation vs the level without laser stimulation and visual stimulation.”

Answer 32: Unfortunately, we cannot perform the exact analysis suggested by the Reviewer. This is because in our experimental design the laser only came on with the stimulus and was off during the blank. We regret this now! However, we have performed an analysis of the firing rates with and without laser at the smallest grating diameters used for each cell, which did not evoke response from the neurons. We found that for most cells the laser induced only very small, and insignificant, changes in firing rate. We have added this analysis to the manuscript (please see lines 184-188).

“3. Approximately 61% (40/66) of single units were significantly modulated by the laser (see Methods).

The numbers were hard to follow: In Fig 2 there were $N = 25$ cells that showed increase in RF size (Fig. 2 b1), $N = 12$ that had increase in RF size and peak response (Fig. 2 b2), and all cells ($N = 33$) in Fig. 2 b3. Additionally there were cells ($N = 12$) showing general suppression (Fig. 3). Maybe the categories are not mutually exclusive? But at first glance this adds up to 49 ($25 + 12 + 12$) or 45 ($33 + 12$). The authors would help the reader a lot by making the numbers clearer.

Answer 32: The Reviewer is correct in that the categories were not mutually exclusive. We have now clarified how the numbers should add up (see Methods lines 527-535 and line 547-553).

“4. The PSTH’s in Fig 2 have the response starting at the onset of the stimulus. Is this a consequence of the smoothing filter? Either change the filter or explain this, otherwise readers will think there is something amiss here. There is always a visual delay.”

Answer 33: Yes, this is a consequence of the smoothing filter. We now explain this in the legend of Fig 2.

“It would also seem to be the case that an analysis of the PSTH dynamics could give further insights into the mechanisms.”

Answer 34: We agree with the Reviewer that an analysis of the PSTH dynamics would give further insights into the mechanisms. However, we prefer not to perform such analysis because the experiment (namely the visual stimulation parameters) was not optimized for this analysis and may, therefore, not produce reliable results.

“5. Analysis of classical RF diameter, the proximal surround size and surround field size (suppl results). There are no details of how these were estimated. It would enhance the paper to either give the methods (were the responses fitted with a model and parameters extracted) or give references to the methods if the method is already published.”

Answer 34: We have added a section to the Methods in which the details of how these parameters were extracted is now explained. We have also performed alternative methods to extract these parameters based on curve fits as suggested by other Reviewers. See above Answers 10 and 22 to other Reviewers.

“6. A couple of minor notes relating to the controls show in the Supplementary data. In Fig.1a top row, 2nd and 3rd columns the laser condition has responses greater than the no laser condition for all

diameters. Yet the summary plot Fig. 1c shows no points above the unity line. It would be helpful to say (as indicated in 5 above) how the peak diameter or proximal surround position were determine for each curve so readers could work out how to go from the curves in Fig 1a to the summary in Fig. 1c.”

Answer 35: Admittedly, this analysis could have been better illustrated and explained. In supplementary Fig. 1b-c of the previous version (now Supplementary Fig. 2c-d) we plot the response with and without laser only at irradiances of 19mW/mm² for supra-granular cells and of 43mW/mm² for granular and infra-granular cells. Therefore, the two example cells the Reviewer is referring to (Fig. 2a top row, 2nd and 3d column), indeed, are excluded from these plots. To increase clarity, we now report the analysis at different irradiances, namely at 43mW/mm² for all cells (Supplementary Fig. 2b), as well as for 19mW/mm² irradiance (for SG cells) and 43mW/mm² (for G and IG cells) (Supplementary Fig. 2c-d). We have also clarified explanation of these control results in the Supplementary Results (see lines 20-51 of Supplementary Results).

“Also the responses in this data set are referred to as responses from “contacts”. Does this mean this is multiunit data? The rates are substantially higher than for the data shown in the main figures. If so, it would be important to indicate that the data is multiunit.”

Answer 36: Yes, these are multi-units. This is now explicitly stated (see lines 18-19 of Supplementary Results).

“Minor points:

1. “In contrast, the responses (no-laser vs. laser: 20.9±8.71 vs. 19.79 ±7.69 spikes/s; mean spike-rate increase 7.10±13.4%, p=0.92).”

Do you mean “spike-rate decrease” here?”

Answer 37: Yes, this has been corrected.

“2. In Suppl doc:

“This is also suggested by modeling the effects of feedback inactivation (Fig. 4b).”
There doesn't seem to be a Fig 4 or any modeling in the manuscript.”

Answer 38: The reference to Fig. 4b has been removed, and models have now been added (see above).

Reviewers' comments:

Reviewer #1 (Remarks to the Author):

In response to the reviewers' concerns, the authors have added additional modeling and data analysis. The paper is now more informative and contains more raw data, but it is not clear how well the overall conclusions align with the data.

My overall impressions are,

- (1) There is on average a change in the grating diameter tuning curves when the V2 feedback is blocked such that the peak ends up occurring for larger size stimuli.
- (2) However, there are many changes to the tuning curves and raw responses, and overall there is no simple way to interpret these changes. Rather than trying to provide a complete report of the variety of changes, the paper leans toward a prior idea that this relates to V2 surround suppression.
- (3) The abstract does not align well with the paper because it is vague and does not clarify what the paper shows versus which popular ideas are plausibly consistent with the findings.
- (4) The neural model does not appear to describe the data very well, before or after the laser perturbation.
- (5) The paper is confusing in places and could be improved by further refinement and input from outside readers.

Specific comments:

The abstract does not directly state the main findings. The findings appear to be that blocking activity in V2 axons into V1 simultaneous with the presentation of a visual stimulus causes a set of diverse changes in response to drifting gratings. This includes suppression of responses to small stimuli, overall enhancement of responses, and frequently an increase in the grating diameter associated with the largest response. How this relates to surround suppression, RF size and response gain are more complicated matters. For example, "RF size" and the diameter of a stimulus that maximally drives a neuron are not generally the same.

The abstract claim to "have identified the cellular mechanisms of feedback-mediated modulations of early sensory processing" is overly broad and vague - I am unsure what "cellular mechanism" has been identified here. The next sentence, "Specifically, ...", does not give any specific cellular mechanism, but lists physiological metrics that change when a perturbation is applied to the system.

The abstract final sentence makes claims that do not follow from the actual data: (1) We do not know if higher cortical areas dynamically regulate spatial resolution based on the experiment in this paper. (2) We do not know if higher cortical areas dynamically regulate sensitivity to image features based on the experiment in this paper. (3) We do not know if higher cortical areas dynamically regulate the efficiency of coding natural images in lower-order areas based on the experiment in this paper.

Line 158-163. The long sentence spanning these lines appears to have a problem because the selection criterion reinforces the result. The result appears to be that the response is larger with laser on vs. laser off, but at a point that was picked because it was the largest among the points on the laser-on tuning curve.

The figure legend for Figure 3a1 is unclear, where it says, "stimuli involving the RF and proximal surround." Does this mean stimuli at the peak of the laser on curve, or stimuli over a range of sizes? The next sentence mentions, "stimuli extending into the proximal surround." Are these the same set of stimulus conditions?

The intention of the term "gain" change is unclear to me. It looks clear that the laser perturbation does not cause a gain change, but rather a change in firing rates that depends in complex ways on the stimulus condition. For example, the response plotted against time in Figure 4 is completely reshaped, and the tuning curve in 4a appears to be cut into on one side. These do not appear to be simply scaled, nor consistent with a scaling of a simple center or surround component.

Line 224 ends without providing an overall message, rather, it reports statistics. It is unclear what conclusions arise from the short section "V2 Feedback Affects Response Gain".

Labels in Figure 4 were confusing. Figure 4c shows that most cells had a higher response for low laser (relative to high laser). So I would expect if you divided low over high, you would get mostly numbers larger than 1, but Figure 4e shows most numbers are lower than 1. Also, 4d shows larger RFs for high laser, so if you divide low-laser over high-laser, I would expect mainly numbers less than 1, but Fig 4e shows RF size changes mostly great than 1.

I had two impressions from the modeling section. First, the model being used in Figure 5 does not produce curves that look like typical V1 summation curves. The model response generally increases with grating diameter except for a ripple in the middle. Second, the changes in the model curves for the "no-FB" cases do not look a lot like the tuning curves shown for the example cells with laser perturbation.

In examples in Fig 2c, 2d, 4a and 4b, the laser condition shows size tuning curves that have large fractional losses of response mainly for small stimuli. Could this be a more striking and consistent result than the change in peaks demonstrated by the examples in 2a, of which one appears to depend largely on a single, noisy point on a tuning curve?

Reviewer #2 (Remarks to the Author):

The authors have addressed my previous comments

Reviewer #3 (Remarks to the Author):

The authors made major changes to the paper and now include modeling that was suggested by three of the reviewers. The modeling gives a clearer understanding of possible mechanisms, allows the results to be compared to other studies that have investigated suppression and provides insights into the changes that might underlie the changes in spatial summation seen with V2 inactivation.

1. Comparing model results to the raw data with respect to receptive field extent.

Answer 27: We have now performed phenomenological model fitting and network model simulations to get better insights into potential mechanisms (see Answers 4 and 23 above).

It appears from the data in Suppl Fig. 5c, using the wc/gc model, that the change in RF size (wc) with

inactivation of feedback would be about a 16% expansion and not the 56% expansion, reported when extracting the values from the raw summation data. Many of the points in suppl Fig. 5c lie on or near the unity line indicating no change in w_c . The change in the value of the 95% point on the spatial summation curve is 56% but the interpretation from the fitting is that this change is predominantly due to changes in other parameters and not a change in the underlying size of the classical receptive field (w_c). The authors need to clarify their interpretation and this partly relates to the next point about defining receptive field size.

The current manuscript goes on to say "Instead, we found that a model involving changes in the spatial extent and gain of the excitatory mechanism best accounted for the range of feedback inactivation effects, suggesting that V2 feedback affects the spatial extent over which cells integrate excitation as well as the gain of excitation."

As pointed out above this is the case for a small minority of the population, most neurons did not appear to change their w_c , at least that is what the results in Suppl. Fig 5c imply. If there is a small but significant change in w_c then this could be determined by using a bootstrap method on a cell-by-cell basis to test the significance of the changes in w_c and the gain.

2. What is meant by receptive field size?

The authors use the stimulus diameter where the summation-tuning curve is maximum or the fitted curves reaches 95% of its maximum as their indicator of receptive field size. They reference Cavanaugh et al (2002) as the source for this measurement as a valid descriptor of the RF size. However, as I read the extract (below) taken from summary to the Cavanaugh et al paper they (Cavanaugh et al) explicitly state that it is the extent of the center mechanism in their model that they used to indicate the size of the classical receptive field. Their interpretation of the point on the spatial summation curve where it reaches 95% of the maximum is that it is due to both the size of the CRF, the surround size and their relative gains, and when this indicator changes with contrast it is due to the change in relative gain not the size. They end up calling this point the "apparent receptive field size". Now the authors have used models that define the receptive field center size do they want to continue to use this point as a measure of receptive field size?

From Cavanaugh et al 2002: "We then develop a receptive field model based on the ratio of signals from Gaussian-shaped center and surround mechanisms. We show that this model can account well for the variations in receptive field size with contrast that we and others have observed and for variations in size with the state of contrast adaptation. The model achieves this success by simple variations in the relative gain of the two component mechanisms of the receptive field. This model thus offers a parsimonious explanation of a variety of phenomena involving changes in apparent receptive field size and accounts for these phenomena purely in terms of two receptive field mechanisms that do not themselves change in size. We used the extent of the center mechanism in our model as an indicator of the spatial extent of the central excitatory portion of the receptive field."

In the current paper the authors need to either justify their use of the peak of the spatial summation curve as a valid measure of receptive field size or maybe use the value of the receptive field center that is returned in their model fitting.

3. A lot of the paper concerns changes in what the authors term the proximal surround. However, as

outlined above the proximal surround depends on where the summation region reaches a maximum, what Cavanaugh et al call the "apparent RF size". The implication from changes seen during laser stimulation is that there is a change in the size of the mechanisms. However, it appears that many of the effects could be accounted for by a change in relative gain of excitatory and inhibitory summation regions along with relatively small changes in sizes of these two regions. Then there is the issue of the additional suppression with large diameters. As pointed out in the discussion and in #4 below there appears to be an additional suppressive influence that comes into play at far from the excitatory center.

4. If there is a version of the DOG model and or the network model that can accommodate the changes in the "apparent proximal surround" then: can you say that V2 accounts for a proportion of the proximal suppression and almost none at large diameters? In addition to showing there is no change at large diameters (such as Fig 3a3), it would be very informative to show the proportion of suppression that is accounted for by V2 (or the model including V2 feedback). In this manner the amount of overall suppression captured by current understanding of V2 effects could be highlighted. Presumably, even though there is a proportion of the attenuation that is seen in the apparent proximal surround due to V2, the majority still comes from V1 or is even precortical. So a more complete discussion could involve showing how much of suppression is due to V2, in addition to documenting the % change from the full suppression condition.

REPLY TO THE REVIEWERS

We thank the Reviewers for their thoughtful comments and constructive criticism and the Editor for the opportunity to revise our manuscript a second time. We have revised the manuscript according to the comments of Reviewers 1 and 3. To facilitate the review of the revised version, the changed text in the revised manuscript is in red font. The following is our point-by-point reply to the Reviewers, with our answers in red text.

REVIEWER 1

“In response to the reviewers' concerns, the authors have added additional modeling and data analysis. The paper is now more informative.....data.

My overall impressions are,

(1) There is on average a change in the grating diameter tuning curves when the V2 feedback is blocked such that the peak ends up occurring for larger size stimuli.”

Answer 1: That is correct.

“(2) However, there are many changes to the tuning curves and raw responses, and overall there is no simple way to interpret these changes. Rather than trying to provide a complete report of the variety of changes, the paper leans toward a prior idea that this relates to V2 surround suppression”.

Answer 2: We feel that in the previous revision we have reported and quantified all of these additional changes the Reviewer is referring to. Specifically, in addition to changes in the size of the summation receptive field (RF), we found significant changes in response amplitude that differed depending on the stimulus size: responses to small stimuli in the RF were significantly reduced, responses to mid-size stimuli extending into the proximal surround were significantly increased (therefore proximal surround suppression was reduced), and responses to large stimuli extending into the distal surround were not significantly changed. All of this information is quantified in Fig. 3, and the corresponding numerical values and statistics are reported in the Results under the section heading **V2 Feedback Affects Response Amplitude in the RF and Proximal Surround**. Moreover, we found that at higher laser intensity, compared to lower laser intensity, most cells showed mean reduced response (the whole size tuning curve was pushed down). These data are reported in Fig. 4. Moreover, contrary to the Reviewer's comment that there is no simple way to interpret all of these changes, we feel that the network model does an excellent job at accounting for most of these changes, using a single mechanism. In particular, when feedback was inactivated both in the model and in our data, responses to stimuli the size of the summation RF were reduced, peak summation was reached at larger stimulus diameters, responses to larger stimuli extending into the proximal surround were increased, and responses to larger stimuli extending into the distal surround were mostly unchanged. Given that the network model was neither designed, nor optimized for explaining any of these effects, but instead was designed to provide insights into the mechanisms underlying the expansion of the summation RF at low stimulus contrast, we think that it is fair to say that the network model provides a

parsimonious, yet powerful explanation of various distinct phenomena in V1, including, but not limited to, the effects of inactivating feedback from V2 to V1.

We agree with the Reviewer that not all the above points were communicated clearly enough in the abstract. We have now modified the abstract (see lines 29-36) as specifically suggested by the Reviewer (see below Answer 6). We have also included in the Results section a more thorough explanation of the various experimental effects of feedback inactivation that the network model reproduces. Please refer to lines 305-306 of the manuscript.

“(3) The abstract does not align well with the paper because it is vague and does not clarify what the paper shows versus which popular ideas are plausibly consistent with the findings”.

Answer 3: We have reworded the abstract as specifically suggested by the Reviewer in the specific comments below (see Answer 6 below). In particular, in the abstract we now report explicitly all of the findings.

“(4) The neural model does not appear to describe the data very well, before or after the laser perturbation.”

Answer 4: The network model is a well-established, well-cited, previously published model of surround suppression. So we are showing nothing new with this model with respect to spatial summation before feedback inactivation. We refer the Reviewer to Answer 15 below for a detailed reply regarding the model. Here we just emphasize that the model, given its simplicity cannot exactly replicate the subtleties and variety of V1 neuron size tuning curves, but it captures the essence of the mechanism. Indeed, it is because of the model’s results in Fig. 5 that we re-analyzed the data and found that the data are in agreement with the model (see Fig. 4).

“(5) The paper is confusing in places and could be improved by further refinement and input from outside readers”.

Answer 5: We have improved writing throughout, as much as we could. The paper has been read by colleagues who are not authors.

“Specific Comments:

The Abstract does not directly state the main findings. The findings appear to be that blocking activity in V2 axons into V1 simultaneous with the presentation of a visual stimulus causes a set of diverse changes in response to drifting gratings. This includes suppression of responses to small stimuli, overall enhancement of responses, and frequently an increase in the grating diameter associated with the largest response.”

Answer 6: We now equally emphasize all of the main findings in the Abstract. The statistically significant findings are: suppression of responses to small stimuli, enhancement of responses in the proximal surround, no response change in the distal surround (the Reviewer refers to an overall response enhancement, but for most cells the enhancement was limited to the proximal surround), and a frequent shift of the peak of the size tuning curve.

“How this relates to surround suppression, RF size and response gain are more complicated matters. For example, "RF size" and the diameter of a stimulus that maximally drives a neuron are not generally the same.”

Answer 7: The diameter of a stimulus that maximally drives a neuron is one definition of RF size (see e.g. Angelucci & Shushruth, 2013), albeit we agree not the only definition of RF size that has been used in the literature. However, this definition has certainly been used in many previous publications, which is why we think it is an equally valid definition of RF size as other definitions. However, to further clarify the terminology used in the paper, and for consistency with previous studies (including our own), we now term this measure of receptive field size the “summation receptive field or sRF” and have replaced RF with sRF in all relevant places in the manuscript.

“The abstract claim to "have identified the cellular mechanisms of feedback-mediated modulations of early sensory processing" is overly broad and vague - I am unsure what "cellular mechanism" has been identified here. The next sentence, "Specifically, ... ", does not give any specific cellular mechanism, but lists physiological metrics that change when a perturbation is applied to the system.”

Answer 8: By cellular mechanisms, we mean the effects of feedback inactivation at the level of single neuron responses. However, we have now removed this terminology from the manuscript.

“The abstract final sentence makes claims that do not follow from the actual data: (1) We do not know if higher cortical areas dynamically regulate spatial resolution based on the experiment in this paper. (2) We do not know if higher cortical areas dynamically regulate sensitivity to image features based on the experiment in this paper. (3) We do not know if higher cortical areas dynamically regulate the efficiency of coding natural images in lower-order areas based on the experiment in this paper.”

Answer 9: This sentence was meant to be speculative. However, it has been removed from the abstract.

“Line 158-163. The long sentence spanning these lines appears to have a problem because the selection criterion reinforces the result. The result appears to be that the response is larger with laser on vs. laser off, but at a point that was picked because it was the largest among the points on the laser-on tuning curve.”

Answer 10: The point of this analysis is to reveal that not only the peak of the size tuning curve was shifted to the right, but in most cells the peak response was of greater amplitude than the control response at the same stimulus size. There is no *a priori* reason to assume that the response at the size tuning peak in the laser-on condition should be higher than the control response. For example, the peak of the size tuning curve in the example cell shown in Figure 2c is shifted to the right, but the response at the peak is lower than the control response. Thus, the finding that the peak response was higher in amplitude than the control response is not enforced by our analysis. In our opinion, this is an important result as, for example, it separates the effects of stimulus contrast and feedback inactivation on size tuning in V1. We now emphasize these

important points more clearly in the manuscript (see page 5 lines 169-173). Moreover, we did not limit our analysis to the peak of the size tuning curve, but also we measured response amplitude at the largest stimulus size used, and found that there was no increase in response amplitude at these largest sizes. Both effects are reported in Fig. 3.

“The figure legend for Figure 3a1 is unclear, where it says, "stimuli involving the RF and proximal surround." Does this mean stimuli at the peak of the laser on curve, or stimuli over a range of sizes? The next sentence mentions, "stimuli extending into the proximal surround." Are these the same set of stimulus conditions?”

Answer 11: Both sentences mean the same, i.e. stimuli at the peak of the laser-on curve. This is now clarified in the legend (it was already clarified in the main text).

“The intention of the term "gain" change is unclear to me. It looks clear that the laser perturbation does not cause a gain change, but rather a change in firing rates that depends in complex ways on the stimulus condition. For example, the response plotted against time in Figure 4 is completely reshaped, and the tuning curve in 4a appears to be cut into on one side. These do not appear to be simply scaled, nor consistent with a scaling of a simple center or surround component.”

Answer 12: The Reviewer is right. We have removed the term gain through the text (including the title) and replaced it with response amplitude.

“Line 224 ends without providing an overall message, rather, it reports statistics. It is unclear what conclusions arise from the short section "V2 Feedback Affects Response Gain".”

Answer 13: We have now added a conclusion sentence to this section (see p.7 lines 236-239).

“Labels in Figure 4 were confusing. Figure 4c shows that most cells had a higher response for low laser (relative to high laser). So I would expect if you divided low over high, you would get mostly numbers larger than 1, but Figure 4e shows most numbers are lower than 1. Also, 4d shows larger RFs for high laser, so if you divide low-laser over high-laser, I would expect mainly numbers less than 1, but Fig 4e shows RF size changes mostly great than 1.”

Answer 14: We are grateful to the Review for pointing out this typo in the axes labels. We have now corrected it by renaming the ratios high over low instead of the original low over high.

“I had two impressions from the modeling section. First, the model being used in Figure 5 does not produce curves that look like typical V1 summation curves. The model response generally increases with grating diameter except for a ripple in the middle. Second, the changes in the model curves for the "no-FB" cases do not look a lot like the tuning curves shown for the example cells with laser perturbation.”

Answer 15: We made a mistake in the original model plots, which generated the false impression of a steep increase in the response at the largest stimuli. We now plot the model size tuning curve as in the original paper by Schwabe et al. (2006). The model represents an average

V1 cell with 30-40% suppression. The “ripple” in the middle is actually an oscillatory component, which in the model is caused by suppression of the excitatory (E) neurons in the near surround driving the center inhibitory (I) neurons. In other words, as the stimulus is expanded beyond the RF, it first activates E neurons in the near surround, which in turn excite the I neurons in the center, leading to surround suppression. However, as the stimulus is further increased in size, the E neurons in the near surround become suppressed, in turn reducing I neurons excitation, thus leading to a release from suppression (a “second rise”). This chain of events carries on with further increases in stimulus size, so the “second rise” is followed by another small suppressive event, which eventually stabilizes. This predicted disinhibition of the center E neurons has been generally overlooked in the experimental data, partly because of the coarser sampling of the stimulus size used in most experiments compared with our computer simulations, and partly because it is generally considered as noise and disregarded by fitting the size tuning curves with phenomenological models. However, this phenomenon has been observed in many models of surround suppression as well as in several experimental papers (DeAngelis *et al.*, 1994; Sengpiel *et al.*, 1997; Walker *et al.*, 2000) (although not commented on by the authors), including our own surround suppression data (e.g. see Fig. 1 below).

Figure 1. Size tuning curves of the model cell (left) and of two example cells recorded in macaque V1 (middle, right). In the middle and right panel, the *black curves* are the empirically measured responses, and the *red curves* the DOG model fits to the data. Note how the model fits fail to capture the “second rise” in the empirical data.

The Reviewer commented that the size-tuning curve with laser-on does not resemble those in the actual data. We agree that the model does not exactly reproduce the shape of the laser-on size-tuning curve in the data. This is expected, due to the simplicity of our model. In particular, the model incorporates feedback connections at a single scale (arising from a single extrastriate area), while real V1 receives feedback inputs at different scales from multiple areas. Therefore, the model cannot optimally reproduce the differential effects on near vs. far surround suppression of removing feedback from a single area, while leaving intact feedback from other areas. Moreover, the model contains only one type of inhibitory neuron as opposed to multiple ones existing in real V1.

However, despite its simplicity, the model captures the main effects of feedback inactivation that we observed experimentally: (i) reduced responses inside the RF (ii) shift of the summation peak to the right (iii) reduced surround suppression. Moreover the model revealed a continuum of effects as a function of reduced feedback activity, which we had initially missed in the data. As feedback activity is progressively reduced (as by increasing laser intensity), RF size

progressively increases and response amplitude progressively decreases. Prompted by these model's results, we analyzed the data and confirmed this model's prediction (Fig. 4). We feel this is a good indication that the model, even if it does not perfectly reproduce the shape of the size tuning curves, is capturing the fundamental mechanisms. Moreover, we also feel that the model's size tuning curves, in fact, do resemble some of the curves in the data: e.g. the cells in Fig. 4a, and 2c (the latter at the largest laser intensity), except that both cells show stronger surround suppression than the model cell.

We feel that adding multiple inhibitory neuron types, feedback at multiple spatial scales, and noise (neuronal response variability) to the model would, indeed, help generate model neurons with size tuning curves that look more similar to the various V1 cells size tuning curves. However, we do not expect to gain significant new knowledge about the underlying feedback mechanisms by performing such a model extension, which would require at a minimum one year of additional work.

“In examples in Fig 2c, 2d, 4a and 4b, the laser condition shows size tuning curves that have large fractional losses of response mainly for small stimuli. Could this be a more striking and consistent result than the change in peaks demonstrated by the examples in 2a, of which one appears to depend largely on a single, noisy point on a tuning curve?”

Answer 16: We feel this result is as significant as the remainder of the results reported in the paper, all of which are described and quantified. That the increase in RF size and response amplitude in the proximal surround are not due to noise has been addressed in our previous revision using two different approaches: 1) fitting the data with DOG or ROG models (Fig. 2d), and 2) performing noise simulations (Supplementary Fig. 4).

REVIEWER 3

“The authors made major changes to the paper and now include modeling that.....V2 inactivation.”

“1. Comparing model results to the raw data with respect to receptive field extent.

Answer 27: We have now performed phenomenological model fitting and network model simulations to get better insights into potential mechanisms (see Answers 4 and 23 above).

It appears from the data in Suppl Fig. 5c, using the wc/gc model, that the change in RF size (wc) with inactivation of feedback would be about a 16% expansion and not the 56% expansion, reported when extracting the values from the raw summation data. Many of the points in suppl Fig. 5c lie on or near the unity line indicating no change in wc. The change in the value of the 95% point on the spatial summation curve is 56% but the interpretation from the fitting is that this change is predominantly due to changes in other parameters and not a change in the underlying size of the classical receptive field (wc). The authors need to clarify their interpretation and this partly relates to the next point about defining receptive field size.”

The current manuscript goes on to say “Instead, we found that a model involving changes in the spatial extent and gain of the excitatory mechanism best accounted for the range of feedback inactivation effects, suggesting that V2 feedback affects the spatial extent over which cells

integrate excitation as well as the gain of excitation.”

As pointed out above this is the case for a small minority of the population, most neurons did not appear to change their wc, at least that is what the results in Suppl. Fig 5c imply. If there is a small but significant change in wc then this could be determined by using a bootstrap method on a cell-by-cell basis to test the significance of the changes in wc and the gain.

Answer 17: The Reviewer is correct in that the change in wc produced by the wc/gc model is indeed smaller than the average 56% change in RF size (defined as the peak of the size tuning curve), because many cells in this model did not change wc. However, the Reviewer should keep in mind that although the wc/gc model performed slightly better than the other 2 parameter models, it only provided the best fit for 30% of the cells. In fact, none of the two-parameter models performed sufficiently well. So, in reality, we feel we cannot reach any strong conclusion as to which parameter best explains the change in RF size. A single parameter model in which only wc is allowed to change (Suppl. Fig 5a top left panel), was actually the best at explaining the change in RF size and reduced response amplitude in the center, but this model could not explain changes in response amplitude in the surround.

So, while we are indeed stating that the wc/gc model provided a somewhat better explanation of the data, we are also stating the following in the Results: “However, none of the two-parameter models provided best fit for the majority of the cells. Moreover, when comparing the different models based on the coefficient of determination (R^2) distributions, rather than fraction of cells best fit by each model, we found that the different models performed similarly (see Supplementary Results and Supplementary Fig. 5d).

“2. What is meant by receptive field size?

The authors use the stimulus diameter where the summation-tuning curve is maximum or the fitted curves reaches 95% of its maximum as their indicator of receptive field size. They reference Cavanaugh et al (2002) as the source for this measurement as a valid descriptor of the RF size. However, as I read the extract (below) taken from summary to Page 5 of 6 the Cavanaugh et al paper they (Cavanaugh et al) explicitly state that it is the extent of the center mechanism in their model that they used to indicate the size of the classical receptive field. Their interpretation of the point on the spatial summation curve where it reaches 95% of the maximum is that it is due to both the size of the CRF, the surround size and their relative gains, and when this indicator changes with contrast it is due to the change in relative gain not the size. They end up calling this point the “apparent receptive field size”. Now the authors have used models that define the receptive field center size do they want to continue to use this point as a measure of receptive field size?

From Cavanaugh et al 2002: “We then.....receptive field.”

In the current paper the authors need to either justify their use of the peak of the spatial summation curve as a valid measure of receptive field size or maybe use the value of the receptive field center that is returned in their model fitting.”

Answer 18: In all our former publications on surround suppression, we have always used the same definitions of RF size as used here, namely the peak of the empirically measured curve or of the fitted curves. This is because we feel that using the Gaussian fit parameters as a measure of the center mechanism relies strongly on the assumption that these models are correct. To the contrary, we do not feel these phenomenological models are excellent descriptors of the data,

particularly the feedback inactivation data in this paper (see Answer 17 above). Moreover, we do not believe these models reveal the actual circuitry or mechanisms underlying center-surround interactions and how these are affected by feedback. For all these reasons, we feel that it is “safer” to report empirical measurements of spatial summation rather than parameters that are strictly bound to a specific model. Finally, as we are ultimately interested in the circuit mechanisms underlying the feedback inactivation effect, we prefer to report the empirical measures of RF size because the latter are the measures we have related to the anatomical spread of long-range connections in V1 in our previous studies (Angelucci *et al.*, 2002; Angelucci & Sainsbury, 2006). We also note that most previous publications have used empirical measures of RF size. However, to further clarify the terminology used in the paper, and for consistency with our previous studies, we now term this measure of receptive field the “summation receptive field or sRF”.

“3. A lot of the paper concerns changes in what the authors term the proximal surround. However, as outlined above the proximal surround depends on where the summation region reaches a maximum, what Cavanaugh et al call the “apparent RF size”. The implication from changes seen during laser stimulation is that there is a change in the size of the mechanisms. However, it appears that many of the effects could be accounted for by a change in relative gain of excitatory and inhibitory summation regions along with relatively small changes in sizes of these two regions.”

Answer 19: Among the phenomenological models, we found that the model in which the gains of the excitatory and inhibitory mechanisms were allowed to change (gc/gs model) performed among the worst (provided best fit only for 2 cells- Suppl Fig. 5b). It is possible that a model in which 4 parameters are allowed to change (gc, gs, wc, ws) would provide much better fits, but then how informative would such a model be? We feel the best insights into the mechanisms of feedback inactivation are provided by the network model, which is why we have relegated the results of the phenomenological models to the Supplementary Material.

“Then there is the issue of the additional suppression with large diameters. As pointed out in the discussion and in #4 below there appears to be an additional suppressive influence that comes into play at far from the excitatory center.

4. If there is a version of the DOG model and or the network model that can accommodate the changes in the “apparent proximal surround” then: can you say that V2 accounts for a proportion of the proximal suppression and almost none at large diameters? In addition to showing there is no change at large diameters (such as Fig 3a3), it would be very informative to show the proportion of suppression that is accounted for by V2 (or the model including V2 feedback). In this manner the amount of overall suppression captured by current understanding of V2 effects could be highlighted. Presumably, even though there is a proportion of the attenuation that is seen in the apparent proximal surround due to V2, the majority still comes from V1 or is even precortical. So a more complete discussion could involve showing how much of suppression is due to V2, in addition to documenting the % change from the full suppression condition.”

Answer 20: This is potentially a very interesting question that could possibly be answered in the future using state-of-the art imaging methods. With the technique used in this study, we unfortunately cannot quantitatively estimate how much our optogenetic manipulation reduced activity in the axons of V2 feedback neurons. However, inactivating thalamocortical axon terminals using eArch3.0 has been shown to reduce, but not to abolish, evoked EPSPs (Mahn *et*

al., 2016), and thus we think that our manipulation only reduced V2 feedback but did not completely abolish it. Moreover, we cannot say to what degree feedback activity was reduced in our study. Hence, we cannot answer this question beyond what we have already done in Fig.3.

LITERATURE CITED

- Angelucci, A, Levitt, JB, Walton, EJ, Hupe, JM, Bullier, J & Lund, JS. (2002). Circuits for local and global signal integration in primary visual cortex. *J Neurosci* **22**, 8633-8646.
- Angelucci, A & Sainsbury, K. (2006). Contribution of feedforward thalamic afferents and corticogeniculate feedback to the spatial summation area of macaque V1 and LGN. *J Comp Neurol* **498**, 330-351.
- Angelucci, A & Shushruth, S. (2013). Beyond the classical receptive field: surround modulation in primary visual cortex. In *The new visual neurosciences*, ed. Chalupa LM & Werner JS, pp. 425-444. MIT press, Cambridge.
- DeAngelis, GC, Freeman, RD & Ohzawa, I. (1994). Length and width tuning of neurons in the cat's primary visual cortex. *J Neurophysiol* **71**, 347-374.
- Mahn, M, Prigge, M, Ron, S, Levy, R & Yizhar, O. (2016). Biophysical constraints of optogenetic inhibition at presynaptic terminals. *Nat Neurosci* **19**, 554-556.
- Sengpiel, F, Sen, A & Blakemore, C. (1997). Characteristics of surround inhibition in cat area 17. *Exp Brain Res* **116**, 216-228.
- Walker, GA, Ohzawa, I & Freeman, RD. (2000). Suppression outside the classical cortical receptive field. *Vis Neurosci* **17**, 369-379.

Reviewers' comments:

Reviewer #3 (Remarks to the Author):

REVIEWER 3

"The authors made major changes to the paper and now include modeling that.....V2 inactivation."

"1. Comparing model results to the raw data with respect to receptive field extent.

Answer 27: We have now performed phenomenological model fitting and network model simulations to get better insights into potential mechanisms (see Answers 4 and 23 above).

It appears from the data in Suppl Fig. 5c, using the wc/gc model, that the change in RF size (wc) with inactivation of feedback would be about a 16% expansion and not the 56% expansion, reported when extracting the values from the raw summation data. Many of the points in suppl Fig. 5c lie on or near the unity line indicating no change in wc. The change in the value of the 95% point on the spatial summation curve is 56% but the interpretation from the fitting is that this change is predominantly due to changes in other parameters and not a change in the underlying size of the classical receptive field (wc). The authors need to clarify their interpretation and this partly relates to the next point about defining receptive field size."

The current manuscript goes on to say "Instead, we found that a model involving changes in the spatial extent and gain of the excitatory mechanism best accounted for the range of feedback inactivation effects, suggesting that V2 feedback affects the spatial extent over which cells integrate excitation as well as the gain of excitation."

As pointed out above this is the case for a small minority of the population, most neurons did not appear to change their wc, at least that is what the results in Suppl. Fig 5c imply. If there is a small but significant change in wc then this could be determined by using a bootstrap method on a cell-by-cell basis to test the significance of the changes in wc and the gain.

Answer 17: The Reviewer is correct in that the change in wc produced by the wc/gc model is indeed smaller than the average 56% change in RF size (defined as the peak of the size tuning curve), because many cells in this model did not change wc. However, the Reviewer should keep in mind that although the wc/gc model performed slightly better than the other 2 parameter models, it only provided the best fit for 30% of the cells. In fact, none of the two-parameter models performed sufficiently well. So, in reality, we feel we cannot reach any strong conclusion as to which parameter best explains the change in RF size. A single parameter model in which only wc is allowed to change (Suppl. Fig 5a top left panel), was actually the best at explaining the change in RF size and reduced response amplitude in the center, but this model could not explain changes in response amplitude in the surround.

So, while we are indeed stating that the wc/gc model provided a somewhat better explanation of the data, we are also stating the following in the Results: "However, none of the two-parameter models provided best fit for the majority of the cells. Moreover, when comparing the different models based on the coefficient of determination (R^2) distributions, rather than fraction of cells best fit by each model, we found that the different models performed similarly (see Supplementary Results and Supplementary Fig. 5d).

Reviewer 3: The receptive field size, now called the summation receptive field (sRF), is defined by the

peak the summation function, but any description of the data would allow this to be estimated. It seems as if the authors conclude that there is little insight to be gained from fitting the phenomenological model (also outlined in their answer 18). This is not apparent in the current manuscript. Changes in combinations of parameters can lead to a change in the peak of the summation function. Combined with the responses to reviewer 1 concerning the complexity of the response profiles it seems as if the authors conclude that the network model is the appropriate way of understanding the mechanisms that give rise to surround modulation. On one hand they include the phenomenological model and yet don't feel it is applicable?

"3. A lot of the paper concerns changes in what the authors term the proximal surround. However, as outlined above the proximal surround depends on where the summation region reaches a maximum, what Cavanaugh et al call the "apparent RF size". The implication from changes seen during laser stimulation is that there is a change in the size of the mechanisms. However, it appears that many of the effects could be accounted for by a change in relative gain of excitatory and inhibitory summation regions along with relatively small changes in sizes of these two regions."

Answer 19: Among the phenomenological models, we found that the model in which the gains of the excitatory and inhibitory mechanisms were allowed to change (gc/gs model) performed among the worst (provided best fit only for 2 cells- Suppl Fig. 5b). It is possible that a model in which 4 parameters are allowed to change (gc, gs, wc, ws) would provide much better fits, but then how informative would such a model be? We feel the best insights into the mechanisms of feedback inactivation are provided by the network model, which is why we have relegated the results of the phenomenological models to the Supplementary Material.

Reviewer 3: If the authors consider the phenomenological models are inadequate in terms of providing insights into how V2 feedback contributes to area summation then maybe they should state this?

4. If there is a version of the DOG model and or the network model that can accommodate the changes in the "apparent proximal surround" then: can you say that V2 accounts for a proportion of the proximal suppression and almost none at large diameters? In addition to showing there is no change at large diameters (such as Fig 3a3), it would be very informative to show the proportion of suppression that is accounted for by V2 (or the model including V2 feedback). In this manner the amount of overall suppression captured by current understanding of V2 effects could be highlighted. Presumably, even though there is a proportion of the attenuation that is seen in the apparent proximal surround due to V2, the majority still comes from V1 or is even precortical. So a more complete discussion could involve showing how much of suppression is due to V2, in addition to documenting the % change from the full suppression condition."

Answer 20: This is potentially a very interesting question that could possibly be answered in the future using state-of-the-art imaging methods. With the technique used in this study, we unfortunately cannot quantitatively estimate how much our optogenetic manipulation reduced activity in the axons of V2 feedback neurons. However, inactivating thalamocortical axon terminals using eArch3.0 has been shown to reduce, but not to abolish, evoked EPSPs (Mahn et al., 2016), and thus we think that our manipulation only reduced V2 feedback but did not completely abolish it. Moreover, we cannot say to what degree feedback activity was reduced in our study. Hence, we cannot answer this question beyond what we have already done in Fig.3.

Reviewer 3: The question didn't relate to data but to the network model. What proportion of the suppression comes from V2 in the model compared to other areas providing feedback? This would give a simple method of matching the experimental results to the model.

REPLY TO THE REVIEWER

We thank the Reviewer for thoughtful comments and constructive criticism. We have revised the manuscript according to the comments of Reviewer 3. To facilitate the review of the revised version, the changed text in the revised manuscript is in red font. The following is our point-by-point reply, with our answers in red text.

REPLY TO REVIEWER 3

-Reviewer 3. The receptive field size, now called the summation receptive field (sRF), is defined by the peak the summation function, but any description of the data would allow this to be estimated. It seems as if the authors conclude that there is little insight to be gained from fitting the phenomenological model (also outlined in their answer 18). This is not apparent in the current manuscript. Changes in combinations of parameters can lead to a change in the peak of the summation function. Combined with the responses to reviewer 1 concerning the complexity of the response profiles it seems as if the authors conclude that the network model is the appropriate way of understanding the mechanisms that give rise to surround modulation. On one hand they include the phenomenological model and yet don't feel it is applicable?

Answer 3. The reason we concluded that the phenomenological models did not provide satisfying insights into the mechanisms underlying the effects of feedback inactivation, was that

the overall performance of the different models was very similar and thus we could not draw a firm conclusion as to which parameter combination would best explain the data. Moreover, all the models perform relatively poorly, being able to fit data for a minority of cells. This is now clearly expressed in the paper (see p. 8 first paragraph, p. 11 end of second paragraph). We included the phenomenological modeling work in the paper as this is what the Reviewers asked us to do. Because we do not feel this is particularly informative, we have moved this part of the modeling study to the Supplementary material, so that readers can be informed that such modeling work was indeed performed but found to be inconclusive.

-Reviewer 3: If the authors consider the phenomenological models are inadequate in terms of providing insights into how V2 feedback contributes to area summation then maybe they show state this?

Answer 4. See Answer 3 above.

-Reviewer 3: The question didn't relate to data but to the network model. What proportion of the suppression comes from V2 in the model compared to other areas providing feedback? This would give a simple method of matching the experimental results to the model.

Answer 5. We cannot address the exact question the Reviewer asked. The model is not intended to replicate all known aspects of cortical circuitry, but rather to show that a single mechanism, i.e. change in excitation-inhibition balance caused by inactivating feedback connections, can qualitatively reproduce most of the findings we have reported in this study. Therefore, the model only includes feedback connections at a single spatial scale, i.e. arising from a single cortical area. Moreover, in the model there is no surround suppression inherited from the LGN. Therefore, while we can estimate surround suppression in the model with intact feedback and with feedback inputs completely removed (the latter leads to a 73% reduction in surround suppression), we believe this percentage does not reflect what happens in the brain, nor our feedback inactivation results. In our experiments, we did not remove all of feedback, but only V2 feedback, and in the real brain, unlike our model, some surround suppression arises in the LGN. However, we were able to perform some data-model comparison (**Fig. 5d-g**), and found that in the model a 50-75% reduction in feedback activity can best reproduce the feedback inactivation data. See also Answer 2 above.

REVIEWERS' COMMENTS:

Reviewer #3 (Remarks to the Author):

The authors have satisfactorily addressed all the remaining questions in the most recent review .